

# Dual-frequency (Ka-band and G-band) radar estimates of liquid water content profiles in shallow clouds

Juan M. Socuellamos[1], Raquel Rodriguez Monje[1], Matthew D. Lebsock[1], Ken B. Cooper[1], and Pavlos Kollias[2,3]

[1]Jet Propulsion Laboratory, California Institute of Technology, Pasadena, CA, USA
[2]Division of Atmospheric Sciences, Stony Brook University, Stony Brook, NY, USA
[3]Department of Environmental and Climate Sciences, Brookhaven National Laboratory, Upton, NY, USA

*Correspondence to*: Matthew D. Lebsock (matthew.d.lebsock@jpl.nasa.gov)

**Abstract.** The profile of the liquid water content (LWC) in clouds provides fundamental information for understanding the internal structure of clouds, their radiative effects, propensity to precipitate, and degree of entrainment and mixing with the surrounding environment. In principle, differential absorption techniques based on coincident dual-frequency radar reflectivity (DFR) observations have the potential to provide the LWC profile. Previous DFR efforts were challenged by the fact that the measurable differential attenuation for small quantities of LWC is usually comparable to the system measurement error. This typically renders the retrieval impractical, as the uncertainty can become many times greater than the retrieved value itself. Theoretically this drawback can be mitigated following two interconnected approaches: (1) increasing the frequency separation between the dual-frequency radar system to measure greater differential attenuation and (2) increasing the radar operating frequency to reduce the instrument measurement random error. Our recently developed 239 GHz radar was deployed at the Eastern Pacific Cloud Aerosol Precipitation Experiment (EPCAPE) along with a variety of collocated remote and in-situ instruments. We have combined Ka-band (35 GHz) and G-band (239 GHz) observations to retrieve the LWC from more than 15000 vertical profiles of shallow clouds with small amounts of LWC. We theoretically and experimentally demonstrate that the pair Ka-band and G-band offers a substantial improvement in the LWC retrieval sensitivity compared to previous works reported in the literature using lower-frequency radars. This new technique provides a missing capability to determine the LWC in the challenging low liquid water path (LWP) range and suggests a way forward to characterize microphysical and dynamical processes more precisely in shallow clouds.

## 1 Introduction

Low-level stratiform clouds have extensive areal coverage over the subtropical oceans, especially near the western continental boundaries (Wood, 2012). These shallow cloud layers moderate the ocean-atmosphere radiative exchange, exerting a strong impact on the sea temperature and the cloud-climate feedback (Brient et al., 2016; Zelinka et al., 2016). Due to their extensive coverage and strong interaction with radiation, shallow clouds can significantly affect the weather on seasonal and shorter time scales (Fast et al., 2019). Understanding the thermodynamic behavior and structure of shallow clouds is therefore



of major importance to improve the representation of the ocean-atmosphere interaction and generate more robust weather forecast models and climate predictions (Bony and Dufresne, 2005; Zelinka et al., 2020).

Given the characteristic shallow thickness of this type of clouds, they are difficult to model in simulations, leading to inaccurate parametrizations that would greatly benefit from additional experimental data (Randall, 2013). A key property that, in large part, describes the radiative impact of shallow clouds is its liquid water content (LWC). The LWC, retrieved at different range gates within the cloud thickness, provides valuable information about the cloud's internal structure, including the turbulent mixing within the cloud layer. Accurate retrievals of LWC in shallow clouds remain, however, a complex endeavor (Ebell et al., 2010). The usual small amounts of liquid water in shallow clouds are hard to measure and require very sensitive instruments and an accurate methodology.

Radiometer and single-frequency radar observations have been used in the past to estimate the LWC relying on assumptions of the vertical LWC profile, the drop size distribution or accurate absolute instrument calibration (Frisch et al., 1998; Lohnert et al., 2001; Matrosov et al., 2004; Illingworth et al., 2007; Kuchler et al., 2018). A more direct technique consists of using a dual-frequency radar system (Atlas, 1954; Eccles and Mueller, 1971; Hogan et al., 2005; Ellis and Vivekanandan, 2011). The estimation of the LWC profiling using the differential frequency radar reflectivity (DFR) technique does not depend on the assumptions made in single-frequency radar approaches. If gaseous attenuation is accounted for, the measured differential attenuation signal from a pair of radar frequencies is proportional to the amount of liquid water in the shallow cloud if drop size effects are negligible (Lhermitte, 1990). In addition, the signal attenuation due to liquid droplets becomes stronger as the operating frequency increases. While the differential technique is not affected by systematic errors, random measurement errors are inversely proportional to the signal frequency. Both facts suggest the use of high frequencies and large frequency pair separations to measure greater differential attenuation and facilitate the retrieval of small LWC values with improved accuracy. This has been demonstrated over the last few decades, as the LWC retrieval accuracy has progressively improved when higher frequency radars became available, with the most recent works using Ka-band (35 GHz) and W-band (94 GHz) frequencies (Hogan et al., 2005; Zhu et al., 2019).

G-band radar instruments are at the new frontier in atmospheric remote-sensing research (Battaglia et al, 2014), offering new possibilities to probe clouds and precipitation and leading to novel measurement techniques. They have been recently used to profile water vapor content in clouds (Cooper et al., 2021) and to provide novel insight into the submillimeter drop size distribution through Doppler measurements (Courtier et al., 2022; Socuellamos et al., 2024a) and multifrequency radar observations (Lamer et al., 2021; Socuellamos et al., 2024b). The synergy of a G-band and a Ka-band radar brings several distinct advantages to the LWC retrieval compared to the W-band and Ka-band pair: (1) increased differential attenuation for submillimeter drop sizes, (2) reduction in the reflectivity measurement random error and, consequently, (3) improved sensitivity in the LWC determination.

The US Department of Energy (DOE) Atmospheric Radiation Measurement (ARM) Mobile Facility (AMF) deployment in the context of the Eastern Pacific Cloud Aerosol Precipitation Experiment (EPCAPE; Russell et al., 2021) offered a convenient opportunity to study shallow clouds through the deployment of several instruments at the Scripps pier in



La Jolla, CA, USA. A ground-based prototype radar called CloudCube participated in EPCAPE for a six-week period in March and April 2023 (Socuellamos et al., 2024b). CloudCube is a multifrequency (Ka-, W- and G-band) profiling radar developed at the Jet Propulsion Laboratory (JPL) under the National Aeronautics and Space Administration Earth Science Technology Office (NASA-ESTO) Instrument Incubator Program (IIP), that includes a 239 GHz (G-band) Doppler module (Socuellamos et al., 2024a). For the analysis developed in this work, we have complemented CloudCube's G-band data with observations

from the ARM Ka-band Zenith-pointing Radar (KAZR; Kollias et al., 2016), as well as ceilometer, microwave radiometer and radiosonde measurements. Using the Ka-band and G-band DFR observations, more than 15000 profiles of LWC have been retrieved for a continuous period of close to 100 minutes, containing different shallow cloud thicknesses, to allow for statistical analysis and validation of the retrieval.

The article starts with a brief description of the radar instruments and the data selected for the analysis. This is

followed by an explanation on the methodology developed to retrieve the LWC and discussion of the results. Vertically integrated LWC is used to validate the results against radiometer liquid water path (LWP) measurements. Finally, we perform an analysis of the adiabaticity of the retrieved LWC profiles and a discussion on the benefits of using G-band frequencies for the LWC retrieval.

## 2. Instruments and data selection

The LWC retrieval described in this article has benefited from the EPCAPE field campaign, which allowed for different instruments to gather and operate simultaneously to enhance the scientific outcomes of the proposed research. The KAZR (Kollias et al., 2016) and CloudCube's G-band radar module (Socuellamos et al., 2024a) are the two main instruments used in this study. For simplicity, CloudCube's G-band module will be abbreviated to simply CloudCubeG in the following. KAZR and CloudCubeG operational parameters during the EPCAPE campaign are described in Table 1.

Besides both radars' data, which form the basis for the LWC retrieval, observational data from other instruments have also been used. In particular, we have worked with ceilometer data to identify the cloud base, and have made use of microwave radiometer (MWR) measurements to validate our retrieval approach by comparing the integrated LWC, i.e. the liquid water path (LWP), with that derived by the radiometer. Additionally, radiosonde data from periodical releases have also been considered to input vertical profiles of temperature, pressure, and relative humidity into our model, and to calculate the gaseous

and liquid attenuation experienced by the radar signals. Figure 1 shows a picture of the location of all the instruments while deployed during the EPCAPE field campaign, also including the location where the radiosondes were released from. The distance between them was not larger than a few tens of meters.




**Table 1: Parameters of KAZR and CloudCubeG instruments during the EPCAPE field campaign.**

|  | KAZR | CloudCubeG |
|---|---|---|
| **Frequency (GHz)** | 34.89 | 238.8 |
| **Transmission type** | Pulsed | FMCW |
| **Pulse width (µs)** | 0.3 | 40 |
| **Pulse repetition interval (ms)** | 0.27 | 0.042 |
| **Peak transmit power (W)** | 100 | 0.24 |
| **Antenna beamwidth (deg)** | 0.19 | 0.35 |
| **Range resolution (m)** | 30 | 10 |
| **Unambiguous range (km)** | 40 | 6.3 |
| **Velocity resolution (ms⁻¹)** | 0.02 | 0.06 |
| **Nyquist velocity (ms⁻¹)** | ±7.97 | ±7.5 |
| **Time resolution (s)** | 4 | 0.4 |

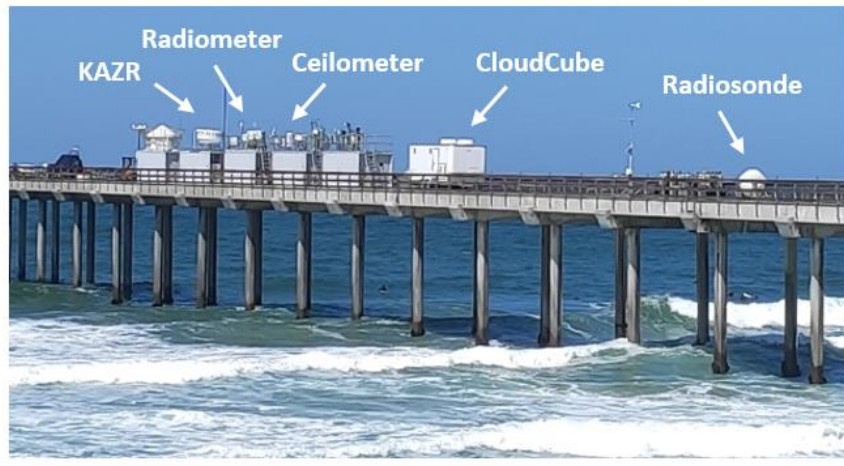


**Figure 1: Instruments during the EPCAPE field campaign at the Scripps pier in La Jolla, CA, USA. The trailers with the KAZR and CloudCubeG instruments were stationed in proximity. The microwave radiometer, ceilometer and the radiosonde release location were also a short distance away.**

Numerous sources of uncertainty in the LWC retrieval have been identified related to the use of the dual-frequency radar technique (Williams and Vivekanandan, 2007), including different radar parameters and sampling mismatches between the radars, presence of ice particles in the cloud volume, potential Mie scattering contamination, and errors in the differential absorption factors used as inputs in the retrieval model. The observations used in this work have been carefully selected based on criteria to mitigate some of these error sources.





The KAZR and CloudCubeG operated with different spatiotemporal resolutions as well as different beamwidths, as shown in Table 1. While these mismatches can contribute to the inaccuracy of the LWC retrieval due to cloud inhomogeneity, we have taken various steps in to minimize these errors. First, we have integrated CloudCubeG range resolution and interpolated KAZR time resolution to obtain data with a common spatiotemporal gate spacing, resulting in 30 m x 0.4 s range-time resolution cells. Second, to mitigate possible errors originating from this artificial matching and the beamwidth mismatch,

we have averaged the data along different range and time gates, as detailed in Sect. 3.1. Averaging over time and range has been proven to improve the errors due to illumination volume mismatch (Hogan et al., 2005; Williams and Vivekanandan, 2007).

     Furthermore, to avoid errors due to the presence of ice particles, the analysis has been focused on full-liquid shallow clouds, identified after studying the radiosonde temperature vertical profile and the G-band radar Doppler spectra. To reduce

Mie scattering contamination, we have excluded regions with precipitating drizzle as deduced from the ceilometer cloud base measurement.

     Finally, the differential gaseous and liquid absorption parameters needed as inputs for the LWC retrieval have been calculated based on in-situ observations from radiosonde data released next to the radars. While the observation selected for the retrieval did not coincide in time with a sonde release, we have compared the data recorded from two sondes that were

released about two hours before and after the selected observation period. The changes in temperature, pressure, and relative humidity between these two sondes were small, giving us confidence that these properties remained relatively stable during the observation period. If the variation was of a small percentage, Hogan et al. (2005), as well as Williams and Vivekanandan (2007), claimed that the propagation error into the LWC retrieval is acceptably small using a Ka-band and W-band pair of radars, and that for more spaced frequencies, the error would further decrease.

The radar data that have been used for the LWC retrieval consist of more than 15000 profiles of observed echo power, comprising a continuous measurement of approximately 100 minutes. The observation was recorded on April 17[th], 2023, at 21:19:00 UTC starting time. The data correspond to a shallow stratocumulus cloud formation with thickness between 120 m and 360 m, containing small amounts of LWC and LWP. This case, particularly challenging for the DFR technique, has been selected to demonstrate the advantages of the Ka-band and G-band proposed retrieval.

The reflectivity data obtained from KAZR and CloudCubeG as well as the mean Doppler velocity data from KAZR are plotted in Fig. 2. The black line describes the cloud base as measured by the ceilometer. Pixels below the cloud base have been removed for the analysis. The reflectivity shown in Fig. 2 has been corrected for gaseous attenuation using the radiosonde data. It is evident when comparing KAZR and CloudCubeG reflectivity plots that the reflectivity is larger at Ka-band than at G-band, and that those differences appear to increase with height above cloud base, which is indicative of the differential

attenuation signal used to derive the LWC profile. There is also some evidence for light drizzle sedimentation as observed by the non-monotonic increase in Ka-band reflectivity with height, especially early in the time series when the cloud is several hundred meters thick. Finally, the Doppler velocity observation shows overturning cells with timescales on the order of 5 to 10 minutes, with a tendency for the scale of the overturning circulations to shorten as the cloud thickness decreases.





**Figure 2: Reflectivity map for KAZR and CloudCubeG and mean Doppler velocity map for KAZR from 21:19:00 UTC to 22:58:00 UTC on April 17th, 2023. The black line represents the cloud base as measured by the ceilometer. Radar data points below the ceilometer baseline have been removed for the analysis. The plots contain more than 15000 reflectivity profiles that have been used to retrieve the LWC throughout 100 minutes of continuous observation.**



## 3. Liquid water content retrieval: methodology and results

### 3.1. Dual frequency reflectivity ratio derivation

  The methodology developed in this article closely follows that proposed in Hogan et al. (2005), and later improved in Zhu et al. (2019), where the LWC retrieval can be performed based on the variation of the DFR across the cloud vertical column. This approach, when performed over a fine range resolution, has been proven to be challenging, as the measurable DFR change is usually comparable to the uncertainty in the radar reflectivity measurement. One possible solution is to average across a few range gates in the vertical dimension and artificially increase the range resolution, so that the DFR between consecutive range gates is increased and the uncertainty reduced. In shallow clouds however, where the cloud thickness is only of a few hundreds of meters, the procedure entails losing valuable information about the variability of the cloud vertical structure. Therefore, we have kept the 30 m original KAZR and integrated CloudCubeG resolution throughout the analysis. This has been possible thanks to the huge leap in wavelengths between KAZR and CloudCubeG, which intensifies the differential attenuation in comparison with shorter-spaced frequency pairs.

  Once most error sources are carefully accounted for, as described in Sect. 2, the main contributor to the uncertainty of the DFR observation is given by the random error of the reflectivity measurements (Hogan et al., 2005). The DFR is simply defined as,

$$DFR = dBZ_K - dBZ_{CCG}, \tag{1}$$

where the subscripts $K$ and $CCG$ refer to KAZR and CloudCubeG, respectively. Then, as derived in Hogan et al. (2005), reflectivity uncertainty can be described in terms of the radar parameters as

$$\Delta dBZ = \frac{10}{\ln 10} \sqrt{\frac{1}{MN}\left(\frac{c}{4\sqrt{\pi}f\sigma_w PRI} + \frac{1}{SNR^2} + \frac{2}{SNR}\right)}, \tag{2}$$

where $c$ is the speed of light, and $f$ and $PRI$ are the frequency and pulse repetition interval of the radars, respectively, with values given in Table 1. The spectrum width ($\sigma_w$) and the signal-to-noise ratio ($SNR$) of the radar signal are complimentary observational data. $N$ is the number of range gates averaged. For CloudCubeG, the range resolution has been increased from 10 m to 30 m to match that of KAZR, therefore $N_{CCG} = 3$. Since no further average in range has been performed, $N_K = 1$. Similarly, $M = t/PRI$ is the number of radar pulses contained within the temporal resolution bin. The temporal resolution of both KAZR and CloudCubeG has been matched at 0.4 s. However, to reduce the random dispersion of the measurements, we set the time step of the data to $t = 60$ s. This number has been selected after carefully analyzing the resulting DFR data for different time steps. While a longer time step reduces the random error, the temporal variability of the LWC retrieval may be affected. A time step of 60 seconds offered an adequate compromise between eliminating evident incorrect artifacts in the data and keeping a relatively fine temporal resolution. In addition, for the rare cases where the difference between two consecutive DFR range bins was greater than 1 dB, presumably as a consequence of the artificial matching of the range resolutions, we applied a spatial filter that replaced the outlying point by a linear interpolation between the preceding and following bins.

  We then calculated the DFR uncertainty as

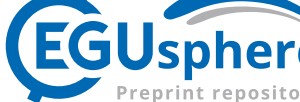


$$\Delta DFR = (\Delta dBZ_K^2 + \Delta dBZ_{CCG}^2)^{1/2}. \tag{3}$$

Both the DFR and associated uncertainty obtained from the observational data, after applying the pertinent time average and range filter, are presented in Fig. 3. The DFR near the cloud base can be seen to have values close to zero, which are expected under the Rayleigh scattering regime, even though these values might be subject to calibration errors (note that for the LWC retrieval, the technique is based on differential DFRs, where calibration errors cancel out). In addition, a DFR of about 3.5 dB at the cloud top can be observed at the beginning of the time window. This is the two-way hydrometeor attenuation in the cloud depth. Following a simplistic approach, we can assume a linear increase of the DFR within the cloud height. In that case, for the range resolution of 30 m, the expected two-way differential attenuation between the Ka and G-band frequencies is in the vicinity of 0.3 dB. For 60 s temporal averaging, this value is larger than the DFR uncertainty error, as can be compared in Fig. 3b. This indicates that the variation of the DFR with range, i.e. $DFR(r_2) - DFR(r_1)$, as needed for the LWC retrieval, could be estimated with a relative error lower than 25% for most the cases. This percentage is however still considerable and suggests the use of additional averaging in the next steps of the LWC retrieval.

Figure 4 shows a single DFR profile at minute 18 of the observation period as pointed out in Fig. 3a. Between 210 m and 270 m, the DFR has a slightly negative increase, which will translate into a negative LWC value if the differential increase is applied directly into the retrieval methodology. Besides the additional averaging previously mentioned, this complication additionally hints at the use of alternative averaging techniques in the following stage.

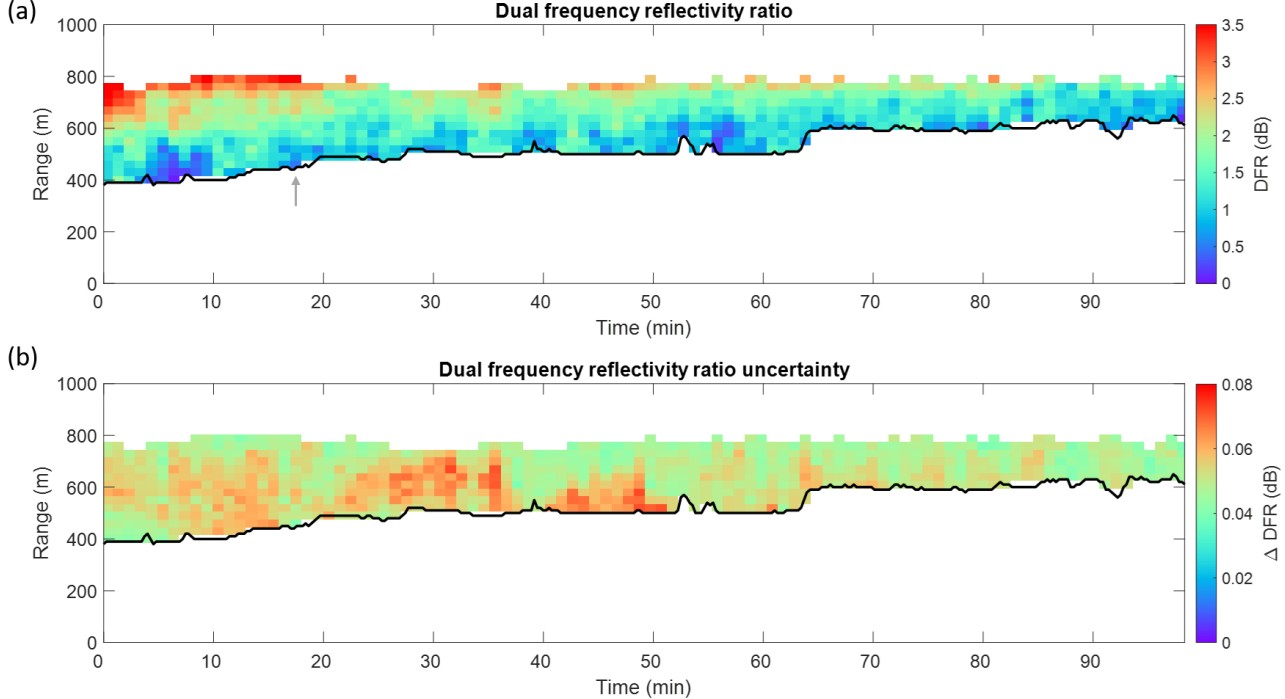

**Figure 3: (a) Dual frequency reflectivity ratio and (b) associated uncertainty for the observation period.**



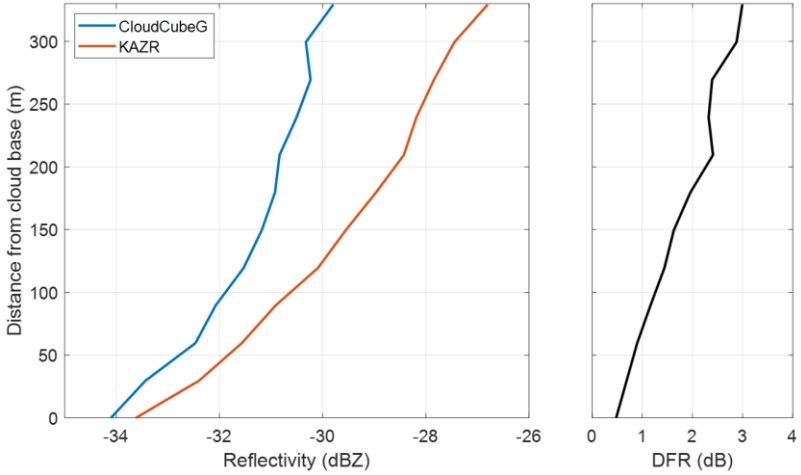

**Figure 4: Reflectivities for CloudCubeG and KAZR (left) and dual frequency reflectivity ratio (right) at minute 18 in Fig. 3.**

### 3.2. Liquid water content retrieval

The possibility to retrieve the LWC using a dual frequency radar system was first discussed in Atlas (1954), and a decade later demonstrated in Eccles and Mueller (1971). These early efforts were focused on rain and convective storms events, envisioning that the LWC could be later used to derive the rainfall rate. Nevertheless, the formulation developed therein is equally valid for low-level shallow clouds, such as the ones discussed in this article. If the radar signals are backscattered under the Rayleigh regime, then the signal attenuation is proportional to the amount of liquid water that the signals penetrate. The LWC at any range gate $r$ can thus be written in terms of the variation of the DFR with range as

$$LWC(r) = \frac{1}{A_l(r)}\left(\frac{\mathrm{d}DFR(r)}{\mathrm{d}r} - A_g(r)\right), \qquad (4)$$

where $A_l$ (in dBkm$^{-1}$g$^{-1}$m$^3$) and $A_g$ (in dBkm$^{-1}$) are the two-way differential liquid and gaseous absorption coefficients, respectively. Based on radiosonde data, we calculated a temperature-dependent $A_l$ following Doviak and Zrnic (1993), and $A_g$ was computed as discussed in Rosenkranz (1998).

The mean LWC in a cloud layer can be calculated by simply replacing the term d$DFR$/d$r$ in Eq. (4) by ($DFR(r_2)$ - $DFR(r_1)$)/ ($r_2$-$r_1$). However, to improve the accuracy of the retrieval, we followed the approach suggested in Zhu et al. (2019), motivated by studies about the LWC profile shape (Küchler et al., 2018). We fit a short profile of $DFR(r)$ data to a second-order polynomial, in such a way that d$DFR$/d$r$ follows a linear relationship with range, that can be described as $2xr + y$, where $x$ and $y$ are the fit parameters. In our case, the short profile of $DFR(r)$ data is composed of 6 vertically consecutive DFR bins, for a group length of 180 m. Starting from the cloud base, the fit is performed along the full vertical extension of the cloud through consecutive sliding profiles that progressively increase the starting height by one range gate. When the starting bin for the profile approaches the cloud top and 6 data points are no longer available, the group length is progressively reduced until





a minimum of 90 m or 3 range gates. Similarly, if the cloud thickness at a given time is shallower than 180 m, the number of bins to group and the fit to perform are adapted accordingly.

Due to the progressive nature of the DFR fitting technique, a different number of short LWC profiles are generated depending on the range position. When more than one LWC profile is available at one range gate, these profiles are averaged to obtain a more accurate LWC. As an example, the first DFR bin at cloud base is only used to derive one LWC profile. The second DFR bin immediately after the cloud base serves as the second data point for the LWC profile starting at cloud base and also as the first data point for the LWC profile starting one range gate after. Therefore, the LWC value at the second range gate is obtained after averaging two LWC short profiles. This compounding effect continues until the group length is achieved and the number of short LWC profiles to average equals that of the number of range gates that form the group.

While this procedure assumes a linear DFR profile shape with height to perform the fit, the fact that the LWC is retrieved after averaging different LWC short profiles that start at consecutive range gates allows for the consideration of the vertical variability of the LWC. The choice of grouping 6 range gates has been determined to not compromise the vertical changes in the LWC. In addition, longer groups had little additional impact on the LWC retrieval accuracy. This range length is similar to the optimal length determined in Zhu et al. (2019).

Hogan et al. (2005), provide an expression to estimate the LWC uncertainty in terms of the DFR random error and the range resolution, $\delta r$. We have adapted the equation to include the number of short LWC profiles averaged ($n$), which improves the accuracy of the final retrieved LWC by $n^{1/2}$, also accounting for the autocorrelation among the distinct LWC short profiles, as

$$\Delta LWC(r) = \frac{1}{\sqrt{n}} \frac{\Delta DFR(r)}{A_l(r)\delta r}. \tag{5}$$

This expression is provided in Hogan et al. (2005), for mean retrievals within a cloud layer and does not consider the additional reduction in the uncertainty due to the fitting technique performed in this work. However, Zhu et al. (2019), estimate based on simulations that the improvement on the LWC accuracy due to the fitting technique could further reduce the uncertainty by a factor 2.

The resulting LWC and uncertainty profiles after completion of the retrieval procedure are shown in Fig. 5 for the observation period. As expected, since the number of short LWC profiles averaged is lower at the cloud base, the uncertainty is greatest there. Then it progressively improves towards the cloud interior, where more averages are performed. We can also see that there are negative LWC values shown in Fig. 5 (which are not physically correct). These values are however mostly located near the cloud base, precisely where the uncertainty is greater. In addition, the negative numbers can become positive within the uncertainty limits. This was one of the criteria that led us to choose the temporal average of 60 s. If the average time is increased to 240 s, the negative values vanish entirely.

At the cloud base, the maximum uncertainty found is 0.22 gm$^{-3}$. At the cloud interior, the uncertainty gets as low as 0.05 gm$^{-3}$. These numbers show that very small amounts of liquid water can be accurately determined deeper into the cloud





boundaries from combining a Ka-band and G-band radar. Moreover, they show a substantial improvement compared to the
previous limits found in Zhu et al. (2019), of 0.10-0.65 gm$^{-3}$ using a Ka-band and W-band radar system.

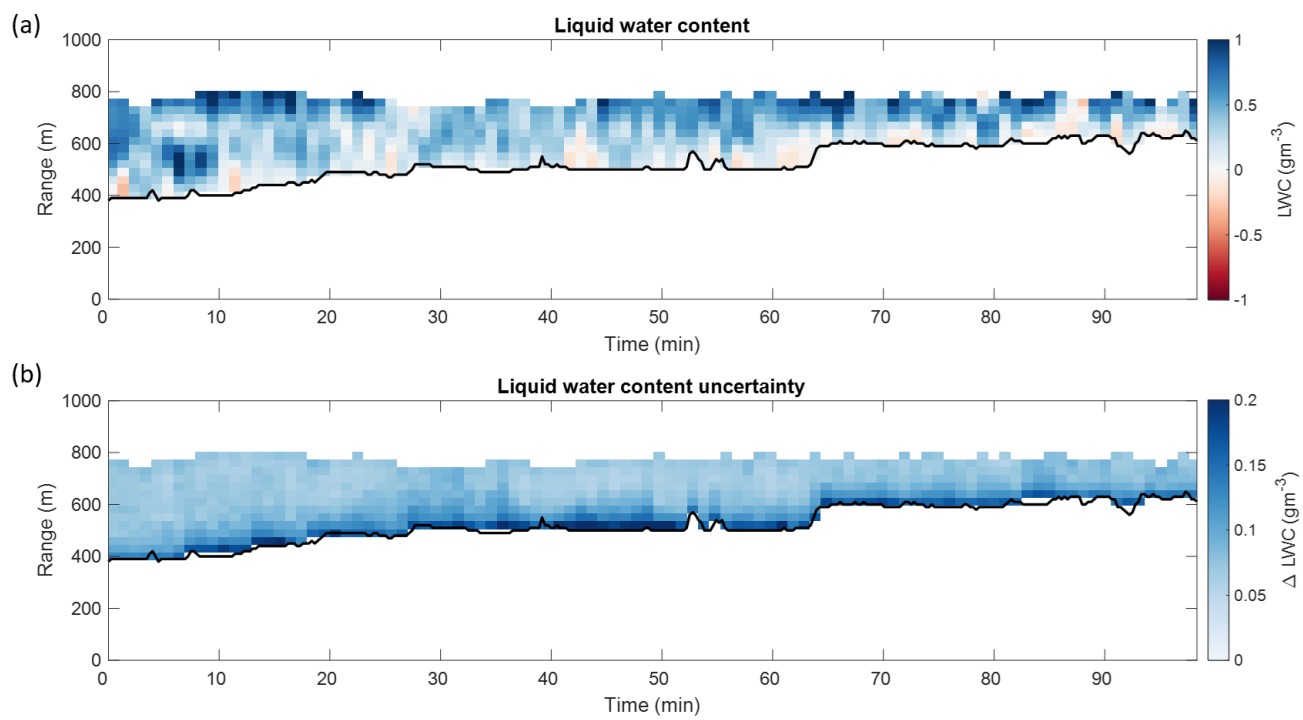

**Figure 5: (a) Retrieved liquid water content and (b) associated uncertainty for the duration of the observation.**

**4. Reflectivity-based retrieval**

        In single-frequency radar systems, the LWC has been traditionally retrieved using an empirical relationship with the
radar reflectivity. Assuming a lognormal distribution of the drop population, this relationship is typically of the form $LWC =
aZ^b$, where $Z$ is the radar reflectivity in linear units and $a$ and $b$ are parameters that describe the drop size distribution (DSD)
in terms of the total drop concentration, $N_o$, and the distribution width, $\sigma$ (Atlas, 1954; Sauvageot and Omar, 1987; Fox and
Illingworth, 1997; Miles et al., 2000; Matrosov et al., 2004). Using a Ka-band radar, the coefficients $a$ and $b$ have been
discussed to be $a = 2.4$ (for maritime clouds) or $a = 4.6$ (for continental clouds) and $b = 0.5$ (Miles et al., 2000; Matrosov et
al., 2004). The values derived in Atlas (1954), Sauvageot and Omar (1987), and Fox and Illingworth (1997), provide similar
results and have not been included in this analysis for simplicity.

        Figure 6 shows the correspondence between KAZR and CloudCube reflectivity values with the retrieved LWC. A
clear trend can be deduced, where the LWC increases for greater values of observed reflectivity. In addition, Fig. 6a includes
the empirical curves for the two previous cases, which show a large systematic low bias of the reflectivity-based approach
compared with the values retrieved from the DFR.





While the parameter $b$ is usually assumed constant and independent of the DSD characteristics, $a$ can adopt a relatively broad range of values depending on $N_o$ and $\sigma$. Using $b = 0.5$ (as in Atlas, 1954, and Miles et al., 2000), $a = (\pi/6)\rho N_o^{0.5}\exp(-4.5\sigma^2)$, where $\rho$ is the liquid water density (Matrosov et al., 2004). The drop concentration and distribution width in Miles et al. (2000) were $N_o = 75$ cm$^{-3}$ (for maritime clouds) or $N_o = 280$ cm$^{-3}$ (for continental clouds) and $\sigma = 0.38$ (for both cases), respectively. A plausible explanation for the discrepancy between the retrieved LWC and the empirical curves may come from the fact that the cloud selected in our observation is formed by a high number of droplets of similar small sizes. This characteristic is typical of continental clouds, which can have drop concentrations close to 1000 cm$^{-3}$ and narrower distribution widths (Squires, 1957; Sassen et al., 1999). Furthermore, recent analysis has shown that observed drop distributions narrow substantially as drop concentration increases (Lebsock and Witte, 2023), which was confirmed in a cloud chamber (Chandrakar et al., 2016). This natural correlation will tend to amplify the dependance of the parameter $a$ on droplet number concentration.

Figure 7 shows the possible range of values that the parameter $a$ can cover for different drop concentrations and distribution widths. The parameter $a$ approaches values of 16 for realizable concentrations of 1000 cm$^{-3}$ with $\sigma < 0.2$.

Figure 8 shows the cloud droplet number concentration derived from the Moderate Resolution Imaging Spectroradiometer (MODIS) aboard the AQUA satellite during an overpass at approximately 20:30:00 UTC. The cloud optical depth and effective radius derived from the 3.7 μm channel are used in the derivation following the approach described in Grosvenor et al. (2018). As seen in Fig. 8, drop concentrations are close to or higher than 400 cm$^{-3}$ near the observation site. Based on this information, we conservatively estimate that droplet concentrations could have been exceeding 300 cm$^{-3}$ during our observation window. For a combination of $N_o > 300$ cm$^{-3}$ and $\sigma < 0.35$, $a$ can become greater than 9. Taking this value as a reference, we can provide a new relationship for the reflectivity-based LWC retrieval

$$LWC = 9Z^{0.5}. \tag{6}$$

The curve obtained by Eq. (6) is also included in Fig. 6a. This relationship provides a better fit with the retrieved LWC, which suggests that the reflectivity-based approach can provide a complimentary result to the DFR technique and potentially constrain the drop concentration in addition to the liquid water content.



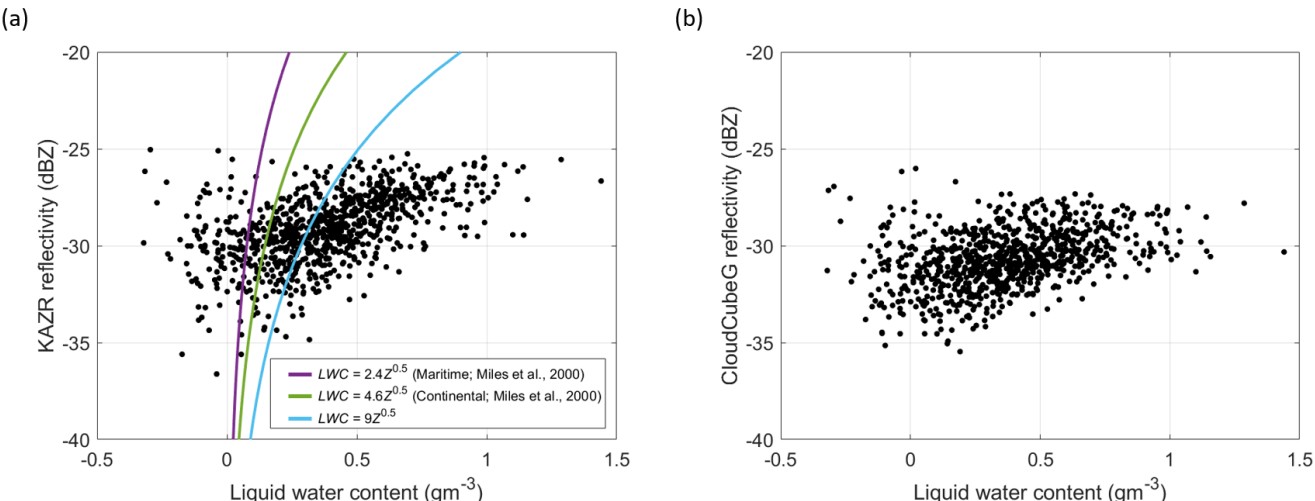

**Figure 6: Comparison of the retrieved liquid water content with (a) KAZR and (b) CloudCube reflectivities. An overall trend can be identified where greater values of LWC correspond to greater reflectivity. Equation (6) provides a better fit to the data than the relationships in Miles et al. (2000).**

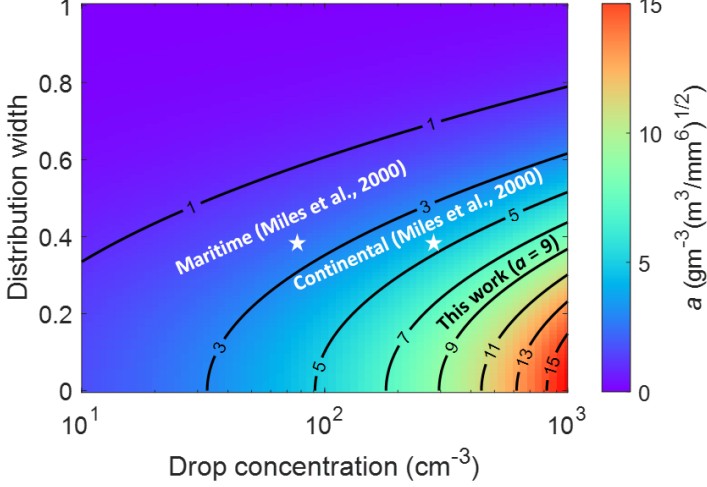

**Figure 7: Values for the parameter $a$ in $LWC = aZ^b$ for different drop concentrations and distribution widths.**





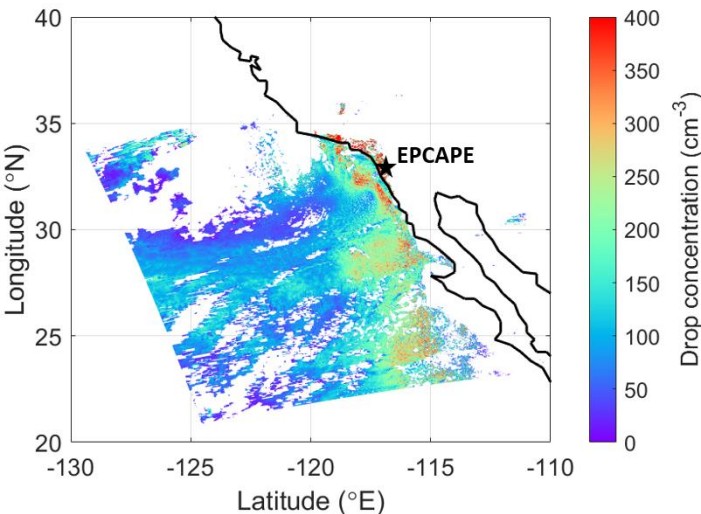

**Figure 8: Drop concentration over the EPCAPE site at approximately 20:30:00 UTC on April 17th, 2023, derived with the method of Grosvenor et al. (2018) using AQUA-MODIS data.**

## 5. Liquid water path retrieval and validation

During the observation period, a microwave radiometer (MWR) at the EPCAPE site recorded the liquid water path (LWP) among other environmental parameters (*mwr3c* data product). The LWP retrieved by the radiometer has been used to validate the LWC retrieval technique by integrating the LWC across the cloud height.

Figure 9a shows the comparison between the radiometer and the radar retrieved LWPs. In addition, we have included the adiabatic LWP, calculated assuming a linear increase of the LWC with range, as explained in Sect. 6. We have also added the LWP following the empirical relationships between the LWC and KAZR reflectivity, as discussed in Sect. 4. All LWP curves have been averaged over 60 s to match the time resolution of the radar LWP. The radar reflectivity-based retrievals based on Miles et al. (2000), are again observed to be clear outliers here with a systematic underestimate of the LWP relative to all other methods. On the other hand, the LWP calculated following Eq. (6) shows excellent correlation with the MWR retrieval. The estimated uncertainty in the MWR derived LWP is 15 gm$^{-2}$ (Cadeddu et al., 2020), while the estimated retrieval uncertainty of the radar DFR LWP is on the order of 10 gm$^{-2}$. The temporal trend of the radiometer LWP is well represented by the DFR radar retrieval, matching the progressive temporal decline of the LWP associated with a shallower cloud towards the end of the observation. For both the radar DFR and radiometer retrieved LWP, the values are generally below the adiabatic curve for approximately the first 65 s of the observation window. Towards the end of the recording period, after 65 s, where the cloud becomes shallower, all three curves tend to match. This is an indication that the thermodynamic behavior of the cloud may be related to its thickness, and is consistent with the observation that the thicker a cloud becomes, the greater its subadiabaticity (Albrecht et al., 1990; Rauber et al., 2007). A comprehensive analysis of the LWC and LWP adiabaticity is performed in Sect. 6.





Figure 9b shows a more detailed comparison of the KAZR-CloudCubeG DFR and the MWR retrieved LWPs. Overall, the relative bias between both retrievals shows large variability along the observation period. The mean bias, represented as a dotted line in Fig. 9b, is -15.9 $gm^{-2}$, which indicates that the KAZR-CloudCubeG retrieved LWP is, on average, greater than

that retrieved by the MWR. This relatively low mean error suggests that a comparison with more temporal averaging will progressively improve the agreement between both retrievals. The root mean square (RMS) deviation, plotted with a dashed line in Fig. 9b, is 33.3 $gm^{-2}$, which is approximately two times larger than what can be explained by the analytic uncertainty estimates in the radar (10 $gm^{-2}$) and radiometer (15 $gm^{-2}$) uncertainties. Given the disperse variability in the sign of the relative bias and the relatively large RMS error, we speculate that there are random sources of uncertainty in the radar retrieval which

have not been accounted for. Two likely causes could be radar-volume mismatches or unidentified influence of drizzle drops on the DFR. However, following a similar validation approach, Hogan et al. (2005) compared the retrieved LWP from a Ka-band and W-band radar system with that from collocated MWRs. For similar amounts of LWP and the same time average as in this work, they found the RMS deviation to be 40 $gm^{-2}$ and 70 $gm^{-2}$ for two different cases and radar systems. While the Ka-band and W-band radars in Hogan et al. (2005) had different beamwidths, as with the KAZR and CloudCubeG here, the radars

in that study were operated with the same range resolution. Furthermore, profiles with Mie scattering were identified corresponding to differential Doppler velocities greater than 0.1 $ms^{-1}$, and excluded from the analysis, which suggests minimal Mie scattering contamination. This speaks for the greater accuracy of the LWC retrieval using a Ka-band and G-band radar system. Even if radar-volume mismatches and Mie scattering contamination may be happening, the RMS deviation shows better agreement between the KAZR-CloudCubeG DFR and the MWR retrieved LWPs than that reported in Hogan et al.

(2005). If in future deployments better care was taken to match the volume sampling of the CloudCubeG and KAZR instruments, we suspect that a significant fraction of the unresolved scatter between the radiometer and the DFR retrieval would be eliminated.



(a)

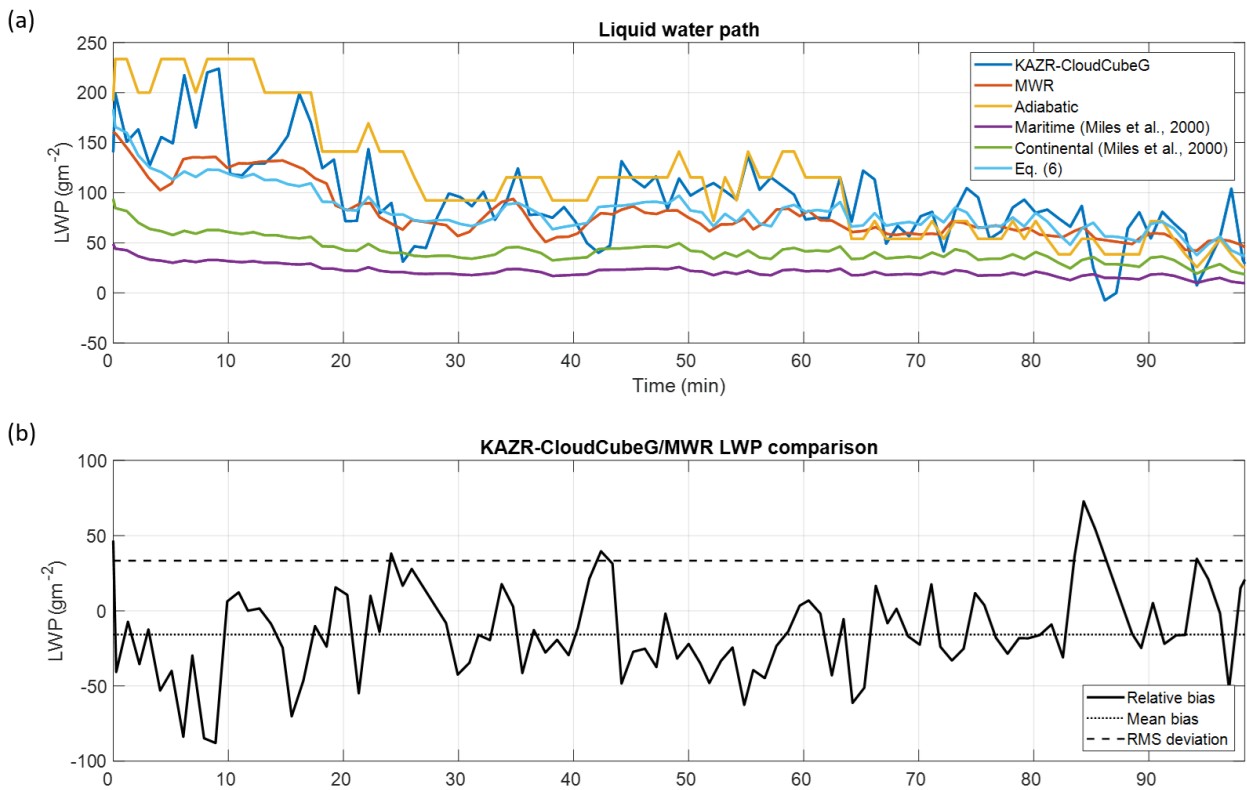

(b)

**Figure 9: (a) Retrieved liquid water path from KAZR and CloudCubeG and comparison with the MWR retrieved LWP and the adiabatic LWP. (b) Comparison between the dual-radar system and the MWR retrieved LWPs, including the relative bias, the mean bias, and the RMS deviation.**

## 6. Liquid water content adiabaticity

To understand the linearity of the LWC retrieval and to provide more insight into the thermodynamic behavior of the shallow cloud, we can compare the results to the case where the LWC increases linearly with range following an adiabatic displacement. In that scenario, the change in the adiabatic liquid water content, $LWC_{ad}$, is driven by the adiabatic lapse rate, $\Gamma_{ad}$, as $LWC_{ad}(r) = \Gamma_{ad}\, r$. Then the ratio between the retrieved LWC and the adiabatic $LWC_{ad}$ describes the adiabatic behavior of the particular cloud layer analyzed, which is defined through the degree of adiabaticity, $f_{ad}$. There are three possibilities: (1) if $f_{ad} = 1$, the LWC in the cloud parcel is exactly the same as if lifted adiabatically within the cloud range step, and therefore there is no exchange of energy with the environment; (2) if $f_{ad} < 1$ the LWC is subadiabatic, indicating than the cloud layer is losing liquid water due to either mixing of warm dry air or sedimentation of drops; and (3) if $f_{ad} > 1$, the LWC is superadiabatic, which can plausibly only be explained by sedimentation of small drizzle drops from above. The degree of adiabaticity is often summarized as a layer mean quantity. In such case, the LWC is replaced by the LWP, and the degree of adiabaticity is equivalently defined as



$$f_{ad} = \frac{LWP}{LWP_{ad}}. \tag{7}$$

To compute $\Gamma_{ad}$, we followed Albrecht et al. (1990), using the temperature, pressure, and humidity measurements of the released radiosondes. $\Gamma_{ad}$ was found to have an average value of 2.85 gm$^{-3}$km$^{-1}$ within the cloud volumes.

Figure 10 shows the LWC and the LWC$_{ad}$ for the different cloud thicknesses measured along the observation period,

which range from 120 m to 360 m. For this representation, we took all the retrieved 60 s profiles that had the same cloud thickness and averaged them. For all the cases except the one corresponding to 300 m, more than 8 profiles were available. The first value at cloud base of the retrieved LWC curves was found, on average, to be biased high by approximately 0.1 gm$^{-3}$, which could correspond to the LWP bias observed in Sect. 5. For a better visual comparison, we have shifted the retrieved LWC profiles in Fig. 10 so that they start exactly at 0 gm$^{-3}$. This shift has not been applied to compute $f_{ad}$.

The degree of adiabaticity based on the LWPs is provided for every case in Fig. 10, and is studied separately in Fig. 11. $f_{ad}$ is greater than one for clouds shallower than 200 m, manifesting a superadiabatic behavior of such clouds. While the uncertainty in the LWC retrieval is larger for shallower cloud sections, which may impact this outcome, this feature agrees with the results reported in Zhu et al. (2019), and has been speculated to be associated to lateral detrainment of cloud water from nearby convective elements (Miller et al., 1998). This analysis is supported by Fig. 12, which shows the mean column

Doppler velocity and the predominant direction of the cloud profiles for the different cloud thicknesses. With the exception of the shallowest cloud section, which has the greatest uncertainty, the superadiabatic behavior is correlated with vertical updrafts, where the mixing of lifted air with humid air near the cloud top may be inducing the sedimentation of cloud droplets. On the other hand, clouds thicker than 200 m show subadiabatic behavior and negative Doppler velocity, suggesting that they sustain a more acute mixing with a drier environment, presumably due to the evaporation of liquid water towards the cloud top, or the

loss of cloud water due to coalescence and sedimentation processes.



**Figure 10: Comparison between the retrieved LWC for different cloud thickness (solid line) and the adiabatic LWC (dashed line). The degree of adiabaticity is displayed for every case. For a better visualization, the retrieved LWC profiles have been shifted to start at zero.**



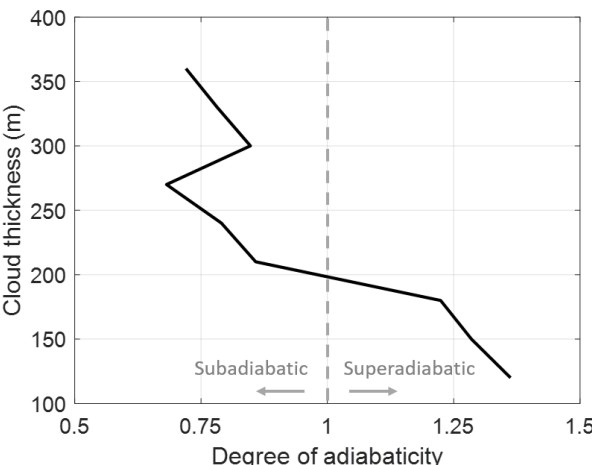

**Figure 11: Degree of adiabaticity as a function of the cloud thickness. Clouds shallower than 200 m show superadiabatic behavior.**

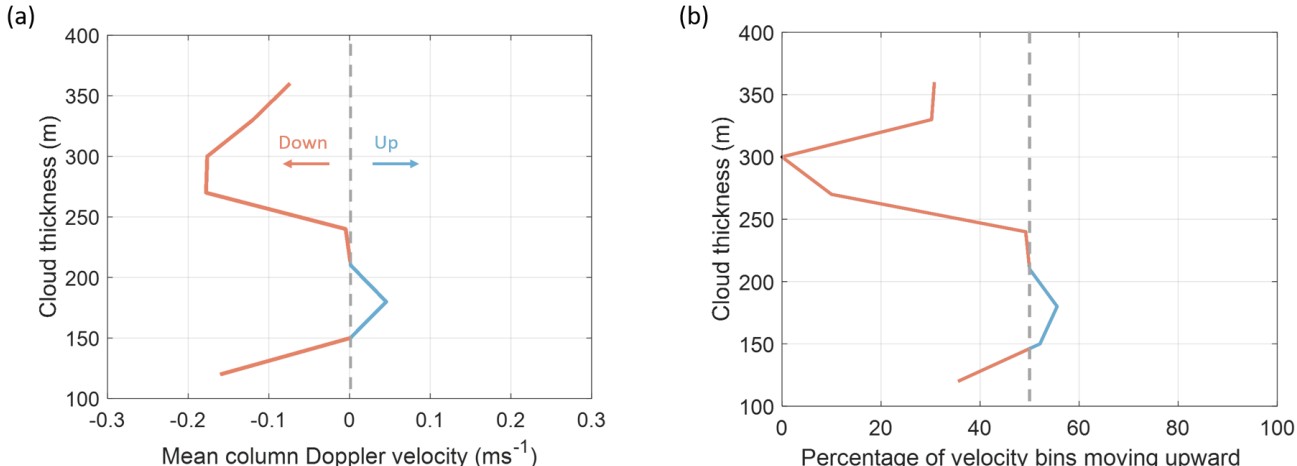

**Figure 12: (a) Mean Doppler velocity across the whole cloud profile for different thicknesses and (b) percentage of Doppler velocity bins that are going up within a cloud profile for different cloud thicknesses.**

### 7. Quantification of the improvement in the LWC retrieval accuracy by using a G-band radar

As described throughout the previous sections, the use of CloudCube's 239 GHz radar has allowed us to retrieve the LWC with improved accuracy in comparison to any other results reported in literature. This is a direct product of the smaller random error in the reflectivity measurement and the use of a complementary lower-frequency radar, with more than 200 GHz separation between both frequencies of operation, that provides increased differential attenuation. To quantify the benefits of using a higher-frequency radar at G-band, we have theoretically compared the LWC uncertainty to a pair of radars where the lower-frequency one operates at either 35 GHz or 94 GHz. To perform this analysis, we have used Eq. (5) keeping the same radar parameters for all the cases and only sweeping the frequency of operation. By taking a closer look at Eq. (5), we can



infer that only the random error in the differential frequency reflectivity ratio, $\Delta DFR$, and the differential liquid attenuation, $A_l$, are frequency dependent parameters. These are precisely the two parameters that will eventually change the LWC retrieval accuracy based on the operating frequency of the radars. To understand their behavior, Fig. 13 shows their relative improvement (meaning a decrease in $\Delta DFR$ and an increase in $A_l$) with respect to the Ka-band and W-band pair, for the two
scenarios aforementioned.

      These two parameters have an opposite effect depending on the lower radar frequency. The relative improvement of the DFR random error is greater if the lower radar frequency is 94 GHz. For the case where the higher frequency radar operates at 239 GHz, as in this work, the partnership with a 35 GHz radar offers an improvement of 8%, while $\Delta DFR$ drops by 37% if it is combined with a 94 GHz radar. This can be understood bearing in mind that, for the same radar parameters, the lower
frequency possesses a greater reflectivity random error that dominates the $\Delta DFR$ term (see Eq. (3)). On the other hand, the differential attenuation between 35 GHz and 239 GHz offers a substantial increase in comparison with the 35 GHz and 94 GHz or 94 GHz and 239 GHz pairs. In particular, the improvement in $A_l$ is about 175% or 2.75 times with the first, and about 60% or 1.6 times with the latter.

      While the use of a higher-frequency radar at 239 GHz clearly offers an improvement in the LWC retrieval accuracy
in regard to the 35 GHz and 94 GHz radar system, whether combining it with a lower-frequency radar at 35 GHz or 94 GHz depends on how $\Delta DFR$ and $A_l$ modify the LWC uncertainty. The results for the LWC after introducing the variation shown in Fig. 13 are presented in Fig. 14. For any of the lower-frequency radars, the LWC retrieval accuracy is always substantially improved if the higher-frequency radar is at 239 GHz compared to the 35 GHz and 94 GHz radar pair. Interestingly, both curves tend to meet as the higher frequency is increased. At 239 GHz, the relative improvement for both cases is around 65%,
meaning that the LWC can be theoretically retrieved to values 3 times smaller in comparison to the Ka-band and W-band radars.

      The proximity of both curves at 239 GHz also indicates that simultaneously retrieving the LWC from a triple-frequency radar system can be beneficial for the LWC retrieval. The Ka-band and W-band pair could be used at regions where the G-band signal suffers from Mie scattering, while combining Ka-band and G-band and W-band and G-band in Rayleigh
scattering locations would provide two independent retrievals that could potentially improve the accuracy by a factor of $2^{1/2}$. Furthermore, the triple-frequency system can be used to identify Mie scattering regions if Doppler is not available, and to facilitate the removal of biases and isolate sporadic instrument artifacts.



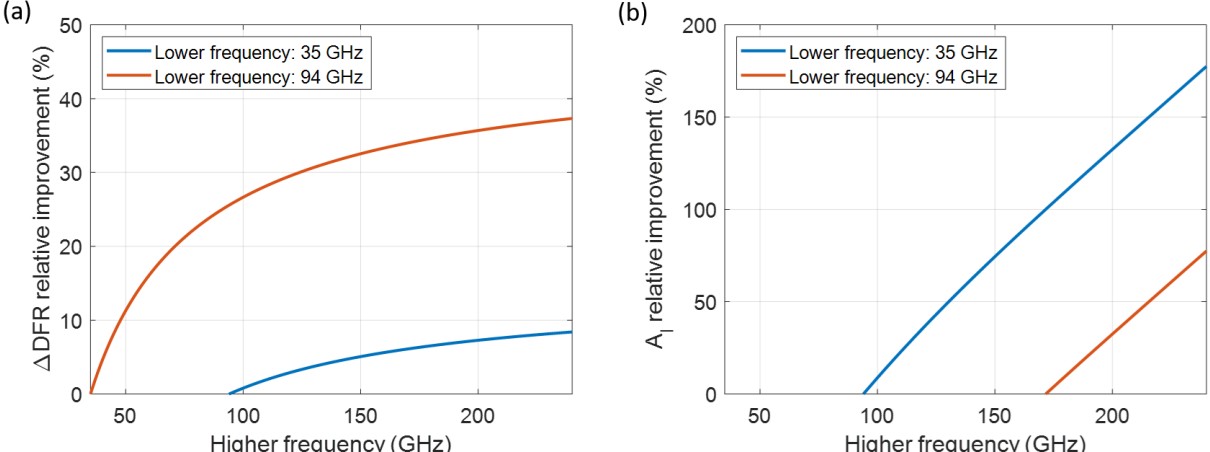

**Figure 13: Relative improvement in the (a) DFR and (b) $A_l$ with respect to the Ka-band and W-band pair for two scenarios: when the higher-frequency radar is complemented with a radar at 35 GHz (blue line) or at 94 GHz (orange line).**

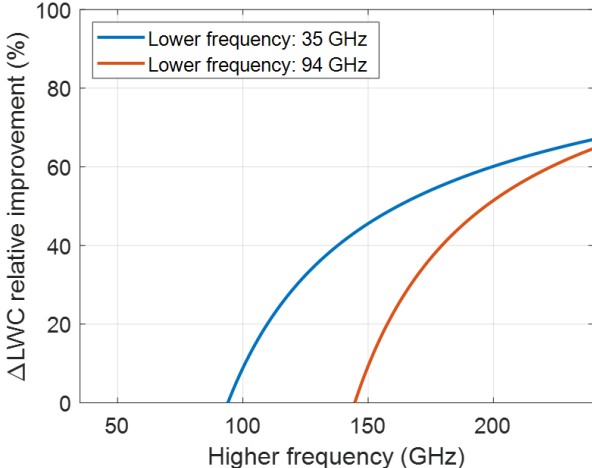

**Figure 14: Relative improvement in the LWC retrieval accuracy with respect to the Ka-band and W-band pair for two scenarios: when the higher-frequency radar is complemented with a radar at 35 GHz (blue line) or at 94 GHz (orange line).**

## 8. Summary and conclusion

Shallow warm clouds have extensive areal coverage over the tropical and subtropical oceans and their strong radiative effects make them an important contributor to Earth's energy balance and hydrological cycle. The vertical profile of the LWC in these warm clouds is indicative of their radiative properties, their ability to precipitate, and the entrainment and mixing with the free tropospheric air.

Single and dual-frequency radar techniques have been previously used to estimate the vertical profile of LWC in these shallow cloud systems. Single-frequency algorithms suffer from assumptions regarding the presence or not of drizzle particles



while dual-frequency algorithms with radar frequencies up to W-band do not have the sensitivity needed to capture the fine structure of the LWC. The dual-frequency radar retrieval accuracy is intrinsically related to the frequencies of operation of the

radar system, where higher frequencies contribute to reducing the measurement error and increasing the liquid water attenuation, then leading to more accurate retrievals. We have demonstrated that the synergy between a 35 GHz and a 239 GHz radar offers distinct advantages to the LWC retrieval accuracy in comparison to a 35 GHz and a 94 GHz radar, eventually allowing measurements of three times smaller liquid water amounts, to LWC values as low as 0.05 gm$^{-3}$ in the inner cloud structure and 0.22 gm$^{-3}$ at the cloud base.

We have accomplished this by deploying the unique 239 GHz radar module of CloudCube ground-based prototype to the EPCAPE field campaign and taking advantage of simultaneous observations of the 35 GHz KAZR and a variety of additional instruments. In total, the analysis included more than 15000 profiles of shallow clouds with small amounts of LWC and LWP, that were selected to demonstrate the benefits of the proposed technique in a particularly challenging scenario. After deriving the dual frequency reflectivity ratio for a 60 s average time to reduce artifacts and errors, we retrieved the LWC across

the cloud thickness after averaging a series of sliding and overlapping LWC short profiles. The DFR technique was able to capture the vertical variability of the LWC, as demonstrated by the comparison of the LWC profiles for different cloud thicknesses with their adiabatic behavior. We found that clouds shallower than 200 m tend to be in a superadiabatic state, while clouds deeper than 200 m behave subadiabatically. On the other hand, we showed how a complimentary reflectivity-based retrieval can be useful to constrain the cloud droplet concentration number in addition to the LWC. We compared the retrieved

LWP to that measured by the collocated MWR. While the direct comparison of 60 s data points showed large variability, the consideration of the mean LWPs showed improved agreement, suggesting that longer average times would lead to better accordance, albeit while negatively affecting the temporal variability of the retrieval. The RMS difference between the MWR and radar DFR derived LWP was 33 gm$^{-2}$, which is roughly twice as large as can be explained by the analytic uncertainty estimates in either retrieval. We speculate that small differences in the sample volume between the G-band and Ka-band radars

are the root cause of this residual uncertainty, and recommend more attention be paid to matching the radar sample volumes in future studies.

The development of G-band radars that are sensitive to smaller characteristic drop sizes is offering new possibilities within the atmospheric remote sensing field. We have proven that G-band radars are a very valuable instrument for the LWC retrieval that, when complemented with a Ka-band or a W-band radar, can potentially determine very small amounts of liquid

water content in low-level shallow clouds. These results can be useful to better understand the vertical structure of shallow clouds and how they interact with the surrounding medium.

**Data availability**

All data sets can be acquired online at https://adc.arm.gov/discovery
- CloudCube: https://doi.org/10.5439/2204401



• KAZR: https://doi.org/10.5439/1498936

       • Ceilometer: https://doi.org/10.5439/1181954

       • Radiometer: https://doi.org/10.5439/1025248

       • Radiosonde: https://doi.org/10.5439/1350631

**Author contributions**

MDL coordinated CloudCube's participation in EPCAPE. RRM managed CloudCube development. RRM, KBC and JMS designed and built CloudCube's G-band radar. JMS, RRM, MDL, and KBC collectively contributed to preparing, installing, and operating CloudCube for and during EPCAPE. JMS processed CloudCube data. JMS and MDL developed the retrieval algorithm. PK provided guidance in the interpretation of the results. JMS composed the manuscript with contributions from the rest of the authors.

**Competing interests**

At least one of the (co-)authors is a member of the editorial board of *Atmospheric Measurement Techniques*.

**Acknowledgements**

This research was supported by the National Aeronautics and Space Administration Earth Science Technology Office (NASA ESTO) under the Instrument Incubator Program, and carried out at the Jet Propulsion Laboratory, California Institute
of Technology, under a contract with NASA (80NM0018D0004).

KAZR, ceilometer, radiometer and radiosonde data were obtained from the atmospheric radiation measurement (ARM) user facility, a U.S. department of energy (DOE) office of science user facility managed by the biological and environmental research program.

© 2024. California Institute of Technology. Government sponsorship acknowledged.

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
