# Peer review of "Dual-frequency (Ka-band and G-band) radar estimates of liquid water content profiles in shallow clouds"

_EGUsphere, 2024_

## Referee Comment (RC1)

The authors demonstrate a differential absorption technique that combines coincident G-band and Ka-band radar observations to obtain vertical profiles of liquid water content (LWC) in shallow clouds. Compared to previous studies using W-band and Ka-band, the new technique results in lower uncertainties due to the increased frequency separation. The improvement due to the use of G-band data is shown both theoretically and from observations. The latter come from about 100 minutes of consecutive measurements of a shallow warm cloud of varying thickness observed during the Eastern Pacific Cloud Aerosol Precipitation Experiment (EPCAPE). The retrieved LWC profiles are compared with single-frequency retrievals as well as microwave radiometer measurements of liquid water path (LWP) and their adiabaticity is analyzed.

The paper is well written and easy to follow. The described technique combining Ka-band and G-band is novel and offers significant advantages over previous efforts in clouds with low LWP. It therefore clearly falls within the scope of AMT. However, I have several concerns that need to be addressed first. I recommend major revision.

**Major comments**

1. It is not clear to me why the Ka- and W-band components from CloudCube are not used in this study. In Sect. 7 the LWC retrieval using Ka-W vs. W-G vs. Ka-G is discussed theoretically. However, I wonder why this was not also investigated using the CloudCube observations? Applying the same retrieval as for Ka-G to Ka-W and W-G and deriving uncertainties should clearly show the improvement using Ka-G band. The authors state in line 436 that a "triple-frequency system can be used to identify Mie scattering regions if Doppler is not available, and to facilitate the removal of biases and isolate sporadic instrument artifacts". I wonder, why is this not done in this study? If the Ka and W-band CloudCube products are not operational yet or had data issues during this period, please say so in the manuscript.

2. I worry that presenting just one case study makes it easy to question the robustness of the results. To me, stating that LWC was retrieved from more than 15000 vertical profiles is questionable, because the retrieval was done for 60 s averages, if I understood correctly. Therefore, LWC was retrieved for 100 profiles, right? Reading the number 15000 feels artificially inflated to me, especially considering that the KAZR data was interpolated. (I want to make clear here that I don't think this was done on purpose out of malicious intent or so.) I recommend including more cases (at least one more) such that the results are more robust. Maybe a case with lower drop concentration according to MODIS would add value to the discussion in Sect. 4?

3. In my opinion, a greater effort to validate the LWC retrieval product not only in terms of column integrated values but also in terms of vertical structure could be taken. The authors could perform (microwave) radiative transfer simulations and forward simulate radar reflectivities given the retrieved amount of liquid water and the assumed drop distribution from Sect. 4. Radar simulators that are flexible in terms of instrument specifications and hydrometeor input are freely available, so I think this could be done without causing too much workload.

4. The authors often use qualitative descriptions, such as "small" or "low" without specifying what they mean quantitatively. This should be clarified in the revision.

5. To my knowledge, figure captions should be limited to describing the figure and only include information necessary to read the figure. Discussion of the figure should be limited to the text. I recommend, the authors have a look at their figure captions again and rewrite where necessary. Also, I suggest writing out any abbreviations used in the figure for readers who only take a look at the figures without reading all of the text.

**Additional comments**

Line 15: Theoretically, …
Line 18: Did you mean "remote sensing"?
Line 20: Please specify what "small" amounts of LWC means in terms of numbers.
Line 23: Please specify what "low" LWP range means.
Line 32: To my knowledge low clouds over the subtropical oceans are one of the largest contributors to uncertainty in climate predictions. This could be stated more explicitly in this paragraph.
Line 33/34: I don't follow the logic here. To me, "difficult to model in simulations" does not necessarily lead to "inaccurate parameterizations". Also "model in simulations" is a bit of a tautology. Maybe you can rephrase this sentence.
Line 34: 2013 is quite a while ago. Are there more recent studies?
Line 35: It should be their, right?
Line 35/36: You have not mentioned remote sensing yet, so I think "range gates" is too specific. I suggest "vertical profile" instead.
Line 38: What do you mean with usual small amounts? Please include typical amounts.
Line 44: Do you mean "profile"?
Line 54: Using the term "new frontier" and citing a ten year old paper feels a bit iffy to me. I am aware (and appreciate) that you give a nice overview of recent studies in the following. Maybe just place the citation differently, e.g. after G-band radar instruments?
Line 66&71: It is not clear to me why you use KAZR instead of the Ka-band component of CloudCube. Please specify.
Line 71: Please specify the temporal resolution.
Line 74: I would recommend you insert references to the corresponding section here.
Line 86/87: If I am not mistaken, you do not specify the instrument model of the used ceilometer and radiometer. Please do so and include references.
Line 88: Please either specify what periodical means or just write "radiosonde data have also been considered".
Figure 1: To me the figure looks rather low quality, especially the white text.
Line 108/109: "based on criteria" sounds very vague, please specify.
Line 112: Remove "in"
Line 115: Since you average your data before performing the LWC retrieval, I don't think you can claim you retrieve LWC from more than 15000 vertical profiles.

Line 120: How did you detect regions with precipitating drizzle from the ceilometer? Please specify.

Line 123: please specify which parameters you use (temperature, relative humidity, pressure?)

Line 125/126: Please quantify what you mean with "small".

Line 127: Again, please quantify what "small percentage" means.

Line 133: Also here (small amounts of LWC and LWP). It's also ok if you specify somewhere (for example in the introduction) that you refer to LWC below xx / LWP below yy as small.

Line 136: Please specify the instrument model and how cloud base height is determined (it is sufficient to state "the internal ceilometer cloud base detection algorithm was used" if that's the case). Also please include the observation frequency, range resolution and uncertainty estimate (probably the range resolution?).

Line 137: I don't think you have mentioned yet, how the correction was done (Rosenkranz?).

Figure 2: I am not a fan of rainbow colormaps and would highly recommend switching to a more color blind friendly colormap (e.g. the python colormaps viridis or plasma). Using a color blindness simulator I found the digital version is not ideal but ok (because the green is lighter than the red tone). However, the printed version of this plot is not readable for people with red-green vision deficiency.

Figure 2: Please mention in the figure caption, if positive or negative doppler velocities mean towards the radar.

Line 182: How rare? Please include percentage values or similar.

Line 192/193: "This is the two-way hydrometeor attenuation in the cloud depth." sounds awkward, can you maybe rephrase?

Line 196: "most of the cases" or "most cases"

Line 198/199: Please specify that you mean 210-270 m above cloud base.

Figure 3 & 5: I suggest you plot the relative uncertainty as well.

Figure 3: Note what the arrow means in the figure caption.

Line 221: "about" sounds awkward to my non-native speaker ears, but maybe that's just me.

Line 221: What does "short" mean? Please be more specific here.

Line 240: What is the optical length determined in Zhu et al. (2019)? Please include it here.

Line 254/255: Please state how many values are 0 within uncertainty ranges and how many negative results remain.

Line 265: Please motivate why you jump from your DFR retrieval to a reflectivity-based approach (for validation? for evaluation which is better? etc)

Line 276: Be specific and state which cases (Miles and Matrosov? Just Miles?)

Line 280-282: "The drop concentration and distribution width in Miles et al. (2000) were $N_o$ = 75 cm$^{-3}$ (for maritime clouds) or $N_o$ = 280 cm$^{-3}$ (for continental clouds) and $\sigma$ = 0.38 (for both cases), respectively." -> awkward sentence, can you rephrase it?

Line 289: Do you mean realistic?

Line 294: I don't get the reasoning. Why don't you use the 400 cm$^{-3}$ from MODIS?

Eq 6: I don't see a good correlation with your retrieval. Is there a physical reason why b must be 0.5?

Line 300: liquid water content -> LWC

Line 314: Same as for the ceilometer. Please specify which instrument was used and which algorithm to retrieve LWP.

Line 322: How much does this depend on the drop concentration? How does it look for 200 or 400 cm$^{-3}$? (Is this the reason why you chose 300 cm$^{-3}$?)

Line 343: To me, the spread in RMS highlights why this study would benefit from more cases.

Figure 9: Can you include uncertainties for MWR and KAZR-CloudCubeG LWP? Maybe also for Eq. 9 using a realistic range of drop concentrations?

Line 360-362: Very long sentence, maybe split in two for better readability?

Line 375/376: What do you mean with the same thickness? Same within +/- 30 m?

Line 377: I know this will be typeset later, but how the unit is split is ugly.

Figure 10: Can you include standard deviation as shaded areas?

Line 405: I suggest adding "to our knowledge"

Line 451/452: Include references here.

Line 456: Where did you demonstrate this? → Add references to the respective figure(s) or section(s).

Line 462: Please state more clearly that the 15000 profiles are from one case study. (Also I'm still not convinced you can claim you use 15000 profiles to retrieve LWC).

Line 470: I suggest you write out abbreviations at least once in the conclusions. Many people will only read abstract and conclusions.

Line 478: The "very" is not necessary, I think "valuable" is convincing enough.

---

## Referee Comment (RC2)

**Review of "Dual-frequency (Ka-band and G-band) radar estimates of liquid water content profiles in shallow clouds" by Socuellamos et al., for publication in Atmospheric Measurement Techniques**

**General Comments**

The novelty of this paper is the combination of Ka-band (35 GHz) and G-band (239 GHz) radar observations for the purpose of retrieving liquid water content (LWC) in very thin clouds during the EPCAPE experiment. 15,000 total vertical profiles of thin cloud were examined in this study, and the retrievals of LWC using this dual-frequency approach demonstrate a convincing improvement compared to previous retrieval efforts using combinations of lower-frequency radar observations. I agree with all of the key results, and especially agree that this approach is a significant advancement even when issues such as Mie scattering or radar volume mismatching are present. Figures 6 and 7 also cleanly show how their Z-LWC relationship fares against previous studies without misleading the readers about the (obvious) spreads in values that are characteristic of such relationships. In addition to this set of important results, this manuscript adds a significant contribution to the realm of G-band radar observations – which will certainly motivate further studies and field campaigns, especially those focused on thin, low-level cloud (which remain a source of high uncertainty in both modeling and observation based studies). The manuscript also does an excellent job of quantifying uncertainty among all used measurements/datasets.

These manuscript qualities make this study an important and significant contribution worthy of publication.

However, I am recommending minor revisions prior to publication for the following reasons, as I believe my comments should not require significant time to address:

The introduction as it stands is well-written but lacking context and relevance to previous works and would benefit from at least another paragraph's worth of context discussing (for example, but not limited to) CloudCube's precursors, previous spaceborne remote sensing cloud retrievals, airborne remote sensing algorithms and related validation studies, and general instrument limitations of previous-generation cloud remote sensing. I was especially surprised – going from the introduction to the methods – the lack of connection (and motivation) to previous spaceborne remote sensing studies using (for example) CloudSat measurements and why CloudCube will be an improvement over a long period of time. I have listed several reading recommendations in my specific comments below – and I encourage the authors to review embedded references within these references to strengthen the introduction further. In addition, I've recommended at several points to either comment on or carry out some additional analysis to better connect your work with the existing literature (the paper itself already has enough publishable results, but I will leave it to the authors to decide the best course of action as I do believe some of these small-but-important details are necessary to strengthen the manuscript).

There are also numerous places in the text (see specific comments below, especially after about ~L120 or so) where the authors need to be more quantitative. In most places, saying "small differences" or "small percentage" (among other similar phrases) is technically correct – but values need to be provided so the reader is not misled in any way. This is especially true for relative humidity or any variable involving moisture.

**Specific/Technical Comments**

L37-39: Sub-cloud evaporation should probably be mentioned here too.

Kalmus, P., & Lebsock, M. (2017). Correcting biased evaporation in CloudSat warm rain. IEEE Transactions on Geoscience and Remote Sensing, 55(11), 6207-6217.

L39: Given this manuscript's connection to CloudCube, it is very important (in my view) to mention limitations of previous satellite-based remote sensing efforts – especially with measuring thin clouds in the lowest 1-km above ground.

Stephens, G. L., et al. (2008), CloudSat mission: Performance and early science after the first year of operation, J. Geophys. Res., 113, D00A18, doi:10.1029/2008JD009982.

L45-47: Can you provide some sort of numeric example explaining what value(s) of differential attenuation correspond to value(s) of LWC? This won't be obvious to your casual reader.

L51: You explain this more in the next sentence, but this statement should be concluded with relevant citations (i.e., what studies "suggest the use of high frequencies and large frequency pair separations?).

L54-61: This is an important statement, but to this point of the introduction, there is limited context for other multi-platform techniques that attempt to quantify cloud liquid (or rain) water. The introduction is especially limited in context from other spaceborne cloud remote sensing studies given this manuscript's connection to CloudCube.

CloudSat + MODIS warm rain retrieval algorithm:

Lebsock, M. D., & L'Ecuyer, T. S. (2011). The retrieval of warm rain from CloudSat. Journal of Geophysical Research: Atmospheres, 116(D20).

L'Ecuyer, T. S., and G. L. Stephens (2002), An estimation-based precipitation retrieval algorithm for attenuating radars, J. Appl. Meteorol., 41, 272–285, doi:10.1175/1520-0450(2002)041<0272:AEBPRA>2.0.CO;2.

Airborne radar & radiometric retrievals of cloud and rain water designed for thin stratocumulus in the SE Atlantic Ocean:

Dzambo, A. M., L'Ecuyer, T., Sinclair, K., van Diedenhoven, B., Gupta, S., McFarquhar, G., O'Brien, J. R., Cairns, B., Wasilewski, A. P., and Alexandrov, M.: Joint cloud water path and rainwater path retrievals from airborne ORACLES observations, Atmos. Chem. Phys., 21, 5513–5532, https://doi.org/10.5194/acp-21-5513-2021

L57: Given the scope of this manuscript, I highly recommend expanding your introduction here to discuss triple-frequency radar observations, and discuss how/why a G-band/Ka-band retrieval is optimal for thin cloud (this will be obvious at this point to most readers, but not everyone will be aware of triple-frequency radar algorithm technology).

Battaglia, A., Tanelli, S., Tridon, F., Kneifel, S., Leinonen, J., & Kollias, P. (2020). Triple-frequency radar retrievals. Satellite Precipitation Measurement: Volume 1, 211-229.

L71-72: To be clear, your study is *not* a statistical analysis of the whole EPCAPE dataset, but rather a single case study based on 100 continuous minutes of data? Please clarify. This information would be more appropriate at the beginning of Section 2.

L90: Consider one of two things in your next version of this paper, either (1) comparing your results (at an appropriate point later in the paper) against those produced by the classic Hitschfeld and Borden (1954) attenuation correction scheme, which is widely used and known, or (2) commenting on how your approach might compare against the classic Hitschfeld and Borden attenuation correction. Most studies that I'm aware of on the topic of radar attenuation use Hitschfeld and Borden – it would be prudent (for visibility sake) that you not neglect this detail.

Hitschfeld, W., & Bordan, J. (1954). Errors inherent in the radar measurement of rainfall at attenuating wavelengths. Journal of the Atmospheric Sciences, 11(1), 58-67.

L118-121: This section can probably be merged with the previous paragraph. Also, "... deduced from the ceilometer cloud base measurement" is somewhat confusing – do you mean to say if the ceilometer detects a cloud base near the surface? I would explain this further.

L126: I presume these "small" differences also accounted for diurnal variability? I would quantify what a "small" difference is in temperature, humidity, etc.

L127: Again, quantify what this "small percentage" is.

L130-134: This information needs to be at the beginning of Section 2, so the reader is clear about what data from what times are used in this analysis. This will also improve the flow of your Section 2 here, as this is awkwardly placed here.

L146: Specify in the caption what positive/negative Doppler velocity corresponds to (i.e., toward or away from the radar).

L159: What is the value of the "uncertainty in the radar reflectivity measurement"? Quantify please.

L164: "huge leap" is a bit informal, and could mean different things to different readers. It may be worth mentioning here (please check in the literature to be sure) that a Ka-band/G-band technique *may* be the widest frequency differential for a technique of this kind (and this will make the novelty of your work more obvious to your reader!).

L221: what exactly is a "short profile"?

L300: You can abbreviate liquid water content to LWC.

L416-423: These are great results – some of these number should be weaved into your abstract to make it more quantitative.

Summary and Conclusions: I have no issues with your algorithm and results and maintain that this work is novel and should be published as soon as possible... however, readers who are familiar with radiometric + radar-based retrievals of cloud LWC may not totally be convinced your work/algorithm is a significant advancement over existing radiometric + radar-based algorithm technology. It would be overkill to add this to your paper, but you should definitely mention in this section as a "future work" that you intend to compare your results against an optimized radiometric + radar joint algorithm to see which one performs better. In particular, a radiometric observation of total optical depth was valuable in constraining the *total* liquid water path of the cloud, and in combination with radar data, can also be used to constrain rainwater contents (see the aforementioned Lebsock and L'Ecuyer, 2011 and Dzambo et al. 2021 references). Again, this would be overkill for this study, but your summary paragraph here should include at least a couple sentences (perhaps a full paragraph) discussing these caveats and avenue of future work.

This section should also include a statement (1-2 sentences) that this work was based on a single set of continuous (100-minute) measurements, and that the robustness of your work will be tested when more data (or similar datasets from other campaigns) are made available.

As an aside – I would really like to see you and your group carry out this suggested follow-on study using radiometric + radar-based retrievals of LWC from EPCAPE!

L449: "indicative of their radiative properties" – LWC is not the *only* property of clouds that matter! Please expand/rephrase to make this more accurate.

L481: I'd say "atmosphere" rather than "medium".

---

## Author Comment (AC1)

Dear Referee,

We would like to thank you for the very useful suggestions to improve the quality of the manuscript. Please see below our response to the comments.

All the changes are highlighted in the revised manuscript.

Sincerely,
The Authors

**General comments**

The authors demonstrate a differential absorption technique that combines coincident G-band and Ka-band radar observations to obtain vertical profiles of liquid water content (LWC) in shallow clouds. Compared to previous studies using W-band and Ka-band, the new technique results in lower uncertainties due to the increased frequency separation. The improvement due to the use of G-band data is shown both theoretically and from observations. The latter come from about 100 minutes of consecutive measurements of a shallow warm cloud of varying thickness observed during the Eastern Pacific Cloud Aerosol Precipitation Experiment (EPCAPE). The retrieved LWC profiles are compared with single-frequency retrievals as well as microwave radiometer measurements of liquid water path (LWP) and their adiabaticity is analyzed.

The paper is well written and easy to follow. The described technique combining Ka-band and G-band is novel and offers significant advantages over previous efforts in clouds with low LWP. It therefore clearly falls within the scope of AMT. However, I have several concerns that need to be addressed first. I recommend major revision.

**Major comments**

1. It is not clear to me why the Ka- and W-band components from CloudCube are not used in this study. In Sect. 7 the LWC retrieval using Ka-W vs. W-G vs. Ka-G is discussed theoretically. However, I wonder why this was not also investigated using the CloudCube observations? Applying the same retrieval as for Ka-G to Ka-W and W-G and deriving uncertainties should clearly show the improvement using Ka-G band. The authors state in line 436 that a "triple-frequency system can be used to identify Mie scattering regions if Doppler is not available, and to facilitate the removal of biases and isolate sporadic instrument artifacts". I wonder, why is this not done in this study? If the Ka and W-band CloudCube products are not operational yet or had data issues during this period, please say so in the manuscript.

We have added the following sentence (line 71-72):

"…the Ka-band and W-band channels of CloudCube did not have the sensitivity needed to detect the cloud formations of interest."

2. I worry that presenting just one case study makes it easy to question the robustness of the results. To me, stating that LWC was retrieved from more than 15000 vertical profiles is questionable, because the retrieval was done for 60 s averages, if I understood correctly. Therefore,

LWC was retrieved for 100 profiles, right? Reading the number 15000 feels artificially inflated to me, especially considering that the KAZR data was interpolated. (I want to make clear here that I don't think this was done on purpose out of malicious intent or so.) I recommend including more cases (at least one more) such that the results are more robust. Maybe a case with lower drop concentration according to MODIS would add value to the discussion in Sect. 4?

We have added an additional case and modified the plots and manuscript accordingly. We have also used the number of averaged LWC profiles throughout the manuscript.

3. In my opinion, a greater effort to validate the LWC retrieval product not only in terms of column integrated values but also in terms of vertical structure could be taken. The authors could perform (microwave) radiative transfer simulations and forward simulate radar reflectivities given the retrieved amount of liquid water and the assumed drop distribution from Sect. 4. Radar simulators that are flexible in terms of instrument specifications and hydrometeor input are freely available, so I think this could be done without causing too much workload.

We intend to perform a more extensive validation of the retrieval technique in a future manuscript once more data are available. We have added the following paragraph at the end of the conclusion (line 492-498):

"Future work will be devoted to testing, improving, and validating the results retrieved in this work. KAZR and CloudCubeG will be participating in the Cloud and Precipitation Experiment at Kennaook (CAPE-k), where close to one-year worth of data will be available by the end of 2025. At CAPE-k, KAZR and CloudCubeG will be accompanied by the W-band ARM Cloud Radar (WACR), and we will exploit the availability of a triple-frequency radar system to benefit the LWC retrieval as described in Sect. 7. Finally, we intend to compare the results against a radiometer and single-frequency radar joint retrieval, as combining the radiometric observation of total optical depth with the radar capability of profiling can also be used to constrain the cloud LWC."

4. The authors often use qualitative descriptions, such as "small" or "low" without specifying what they mean quantitatively. This should be clarified in the revision.

We have addressed this issue in the new version of the manuscript.

5. To my knowledge, figure captions should be limited to describing the figure and only include information necessary to read the figure. Discussion of the figure should be limited to the text. I recommend, the authors have a look at their figure captions again and rewrite where necessary. Also, I suggest writing out any abbreviations used in the figure for readers who only take a look at the figures without reading all of the text.

We have modified figures captions to delete unnecessary information and write out abbreviations.

**Additional comments**

Line 15: Theoretically, …

The recommendation has been accepted (line 15).

Line 18: Did you mean "remote sensing"?

The recommendation has been accepted (line 18).

Line 20: Please specify what "small" amounts of LWC means in terms of numbers.

We have added the following correction in line 20:

"…typical amounts of LWC smaller than 1 $gm^{-3}$."

Line 23: Please specify what "low" LWP range means.

We have added the following correction in line 23:

"…(< 200 $gm^{-2}$)"

Line 32: To my knowledge low clouds over the subtropical oceans are one of the largest contributors to uncertainty in climate predictions. This could be stated more explicitly in this paragraph.

We have added the following sentence in line 29-30:

"Variability in the response of these clouds to anthropogenic warming remains a significant source of spread in climate change projections (Zelinka et al., 2020)."

Line 33/34: I don't follow the logic here. To me, "difficult to model in simulations" does not necessarily lead to "inaccurate parameterizations". Also "model in simulations" is a bit of a tautology. Maybe you can rephrase this sentence.

We have modified the paragraph (line 35-37):

"Given their characteristic thickness, shallow clouds are challenging to accurately represent in global-scale simulations. Incorporating additional experimental data can be of use to develop more appropriate parametrizations for climate models."

Line 34: 2013 is quite a while ago. Are there more recent studies?

We have added two new and more recent references (line 37):

Medeiros, B., Shaw, J., Kay, J. E., and Davis, I.: Assessing clouds using satellite observations through three generations of global atmosphere models. Earth and Space Sci., 10(7), e2023EA002918. https://doi.org/10.1029/2023EA002918, 2023.

Schneider, T., Leung, L. R., and Wills, R. C. J.: Opinion: Optimizing climate models with process knowledge, resolution, and artificial intelligence, Atmos. Chem. Phys., 24, 7041–7062, https://doi.org/10.5194/acp-24-7041-2024, 2024.

Line 35: It should be their, right?

The recommendation has been accepted (line 38).

Line 35/36: You have not mentioned remote sensing yet, so I think "range gates" is too specific. I suggest "vertical profile" instead.

The recommendation has been accepted (line 38).

Line 38: What do you mean with usual small amounts? Please include typical amounts.

We have added the following correction in line 41:

"…($< 1$ gm$^{-3}$)"

Line 44: Do you mean "profile"?

The recommendation has been accepted (line 46).

Line 54: Using the term "new frontier" and citing a ten year old paper feels a bit iffy to me. I am aware (and appreciate) that you give a nice overview of recent studies in the following. Maybe just place the citation differently, e.g. after G-band radar instruments?

We have modified the sentence and added a new and more recent reference (line 56-57):

"G-band radar instruments are a recently-developed capability…"

Battaglia, A., Kollias, P., Dhillon, R., Roy, R., Tanelli, S., Lamer, K., Grecu, M., Lebsock, M., Watters, D., Mroz, K., Heymsfield, G., Li, L., and Furukawa, K.: Spaceborne cloud and precipitation radars: status, challenges, and ways forward, Reviews of Geophysics, 58, https://doi.org/10.1029/2019rg000686, 2020.

Line 66&71: It is not clear to me why you use KAZR instead of the Ka-band component of CloudCube. Please specify.

We have added the following sentence (line 71-72):

"…the Ka-band and W-band channels of CloudCube did not have the sensitivity needed to detect the cloud formations of interest."

Line 71: Please specify the temporal resolution.

We have modified the sentence (line 75):

"…averaged over 60 s"

Line 74: I would recommend you insert references to the corresponding section here.

The recommendation has been accepted (line 76-80).

Line 86/87: If I am not mistaken, you do not specify the instrument model of the used ceilometer and radiometer. Please do so and include references.

We have added the following references in line 73-74:

Cadeddu, M. P.: Microwave Radiometer – 3-Channel (MWR3C) Instrument Handbook. U.S. Department of Energy, Atmospheric Radiation Measurement user facility, Richland, Washington.

DOE/SC-ARM-TR-300, 2024. Available online: https://www.arm.gov/publications/tech_reports/handbooks/doe-sc-arm-tr-300.pdf, last access: September 13, 2024.

Morris, V. R.: Ceilometer Instrument Handbook. U.S. Department of Energy, Atmospheric Radiation Measurement user facility, Richland, Washington. DOE/SC-ARM-TR-020, 2016. Available online: https://www.arm.gov/publications/tech_reports/handbooks/ceil_handbook.pdf, last access: September 13, 2024.

Line 88: Please either specify what periodical means or just write "radiosonde data have also been considered".

The recommendation has been accepted (line 90).

Figure 1: To me the figure looks rather low quality, especially the white text.

We have improved the quality of Fig. 1.

Line 108/109: "based on criteria" sounds very vague, please specify.

We have deleted the sentence.

Line 112: Remove "in"

The recommendation has been accepted.

Line 115: Since you average your data before performing the LWC retrieval, I don't think you can claim you retrieve LWC from more than 15000 vertical profiles.

We have used the number of averaged LWC profiles throughout the manuscript.

Line 120: How did you detect regions with precipitating drizzle from the ceilometer? Please specify.

We have modified the sentence (line 122):

"… we have excluded regions below the cloud base measured by the ceilometer."

Line 123: please specify which parameters you use (temperature, relative humidity, pressure?)

We have added the following correction (line 124):

"…measurements of pressure, temperature, and relative humidity"

Line 125/126: Please quantify what you mean with "small".

We have added the following correction (line 128):

"…less than 0.5 K, 7 hPa, and 3%, respectively"

Line 127: Again, please quantify what "small percentage" means.

We have added the following correction (line 129-131):

"Even if there had been a variation of 1 K, … would be smaller than 0.03 gm$^{-3}$…"

Line 133: Also here (small amounts of LWC and LWP). It's also ok if you specify somewhere (for example in the introduction) that you refer to LWC below xx / LWP below yy as small.

We have added the following correction (line 100):

"…containing characteristic LWC and LWP amounts smaller than 1 gm$^{-3}$ and 200 gm$^{-2}$, respectively."

Line 136: Please specify the instrument model and how cloud base height is determined (it is sufficient to state "the internal ceilometer cloud base detection algorithm was used" if that's the case). Also please include the observation frequency, range resolution and uncertainty estimate (probably the range resolution?).

We have added the following sentence (line 134-135):

"…by the internal ceilometer cloud base detection algorithm (*ceil* data product) with temporal and range solutions of 16 s and 30 m, respectively."

Line 137: I don't think you have mentioned yet, how the correction was done (Rosenkranz?).

We have added the reference (line 137).

Figure 2: I am not a fan of rainbow colormaps and would highly recommend switching to a more color blind friendly colormap (e.g. the python colormaps viridis or plasma). Using a color blindness simulator I found the digital version is not ideal but ok (because the green is lighter than the red tone). However, the printed version of this plot is not readable for people with red-green vision deficiency.

We have used the parula colormap in the new version of the plots.

Figure 2: Please mention in the figure caption, if positive or negative doppler velocities mean towards the radar.

We have added the following sentence (line 147-148):

"Negative Doppler velocities correspond to particles moving towards the radar."

Line 182: How rare? Please include percentage values or similar.

We have added the following correction (line 177):

"…(less than 1% of the total data)"

Line 192/193: "This is the two-way hydrometeor attenuation in the cloud depth." Sounds awkward, can you maybe rephrase?

We have deleted the sentence.

Line 196: "most of the cases" or "most cases"

The recommendation has been accepted (line 192).

Line 198/199: Please specify that you mean 210-270 m above cloud base.

The recommendation has been accepted (line 195).

Figure 3 & 5: I suggest you plot the relative uncertainty as well.

The recommendation has been accepted.

Figure 3: Note what the arrow means in the figure caption.

We have added the following sentence (line 201):

"The arrow at minute 18 in (a) points at the profile used in Fig. 4."

Line 221: "about" sounds awkward to my non-native speaker ears, but maybe that's just me.

We have replaced "about" by "related to" (line 218).

Line 221: What does "short" mean? Please be more specific here.

We have changed the sentence (line 218):

"We fit a portion of the DFR(r) profile…"

Line 240: What is the optical length determined in Zhu et al. (2019)? Please include it here.

We have added the following (line 237):

"…of 187.5 m"

Line 254/255: Please state how many values are 0 within uncertainty ranges and how many negative results remain.

We have added the following (line 252):

"…more than 95% of"

Line 265: Please motivate why you jump from your DFR retrieval to a reflectivity-based approach (for validation? for evaluation which is better? etc)

We have added the following sentence (line 266-267):

"These relationships provide a straightforward approach to estimate the LWC and can also serve as a point of comparison for different techniques."

Line 276: Be specific and state which cases (Miles and Matrosov? Just Miles?)

We have corrected the sentence and added the reference (line 276):

"…for the two cases in Miles et al. (2000)"

Line 280-282: "The drop concentration and distribution width in Miles et al. (2000) were No = 75 cm-3 (for maritime clouds) or No = 280 cm-3 (for continental clouds) and $\sigma$ = 0.38 (for both cases), respectively." -> awkward sentence, can you rephrase it?

We have modified the sentence (line 280-281):

"In Miles et al. (2000), the drop concentration was found to be 75 cm$^{-3}$ for maritime clouds and 280 cm$^{-3}$ for continental clouds, while the distribution width was 0.38 for both cases."

Line 289: Do you mean realistic?

The recommendation has been accepted (line 289).

Line 294: I don't get the reasoning. Why don't you use the 400 cm-3 from MODIS?

Note that the MODIS overpasses did not coincide in time with our observations. We prefer to use a more conservative number for this calculation.

Eq 6: I don't see a good correlation with your retrieval. Is there a physical reason why b must be 0.5?

$b$ is most commonly taken equal to 0.5 because of the proportionality of $Z$ and $LWC$ to different moments of the drop size distribution ($Z$ proportional to $D^6$ and $LWC$ proportional to $D^3$).

Line 300: liquid water content -> LWC

The recommendation has been accepted (line 301).

Line 314: Same as for the ceilometer. Please specify which instrument was used and which algorithm to retrieve LWP.

We have added the data product and the reference (line 321).

Line 322: How much does this depend on the drop concentration? How does it look for 200 or 400 cm$^{-3}$? (Is this the reason why you chose 300 cm$^{-3}$?)

Note that the parameter $a$ in Eq. (6) accounts for a range of drop concentrations > 300 cm$^{-3}$ (and a range of distribution widths).

Line 343: To me, the spread in RMS highlights why this study would benefit from more cases.

We have added one more case.

Figure 9: Can you include uncertainties for MWR and KAZR-CloudCubeG LWP? Maybe also for Eq. 9 using a realistic range of drop concentrations?

The uncertainties for MWR and KAZR-CloudCubeG LWP are around 10 gm$^{-2}$, which would be barely visible in the plot. We prefer not to include uncertainties to provide a cleaner plot.

Line 360-362: Very long sentence, maybe split in two for better readability?

We have corrected the sentence (line 366-367):

"To assess the thermodynamic behavior of the shallow cloud, we can compare the linearity of the LWC retrieval to the adiabatic case where the LWC increases linearly with range."

Line 375/376: What do you mean with the same thickness? Same within +/- 30 m?

We have added the following correction (line 380):

"…in steps of 30 m."

Line 377: I know this will be typeset later, but how the unit is split is ugly.

This has been corrected in the new version.

Figure 10: Can you include standard deviation as shaded areas?

The recommendation has been accepted.

Line 405: I suggest adding "to our knowledge"

The recommendation has been accepted (line 411).

Line 451/452: Include references here.

To our understanding, references should be avoided in the conclusion. Previous works related to this particular sentence are referenced in the introduction (line 44-46).

Line 456: Where did you demonstrate this? → Add references to the respective figure(s) or section(s).

We have added the following correction (line 465):

"…in Sect. 3.2"

Line 462: Please state more clearly that the 15000 profiles are from one case study. (Also I'm still not convinced you can claim you use 15000 profiles to retrieve LWC).

We have used the number of averaged LWC profiles throughout the manuscript.

Line 470: I suggest you write out abbreviations at least once in the conclusions. Many people will only read abstract and conclusions.

The recommendation has been accepted.

Line 478: The "very" is not necessary, I think "valuable" is convincing enough.

The recommendation has been accepted.

---

## Author Comment (AC2)

Dear Referee,

We would like to thank you for the very useful suggestions to improve the quality of the manuscript. Please see below our response to the comments.

All the changes are highlighted in the revised manuscript.

Sincerely,
The Authors

**General Comments**

The novelty of this paper is the combination of Ka-band (35 GHz) and G-band (239 GHz) radar observations for the purpose of retrieving liquid water content (LWC) in very thin clouds during the EPCAPE experiment. 15,000 total vertical profiles of thin cloud were examined in this study, and the retrievals of LWC using this dual-frequency approach demonstrate a convincing improvement compared to previous retrieval efforts using combinations of lower-frequency radar observations. I agree with all of the key results, and especially agree that this approach is a significant advancement even when issues such as Mie scattering or radar volume mismatching are present. Figures 6 and 7 also cleanly show how their Z-LWC relationship fares against previous studies without misleading the readers about the (obvious) spreads in values that are characteristic of such relationships. In addition to this set of important results, this manuscript adds a significant contribution to the realm of G-band radar observations – which will certainly motivate further studies and field campaigns, especially those focused on thin, low-level cloud (which remain a source of high uncertainty in both modeling and observation based studies). The manuscript also does an excellent job of quantifying uncertainty among all used measurements/datasets.

These manuscript qualities make this study an important and significant contribution worthy of publication.

However, I am recommending minor revisions prior to publication for the following reasons, as I believe my comments should not require significant time to address:

The introduction as it stands is well-written but lacking context and relevance to previous works and would benefit from at least another paragraph's worth of context discussing (for example, but not limited to) CloudCube's precursors, previous spaceborne remote sensing cloud retrievals, airborne remote sensing algorithms and related validation studies, and general instrument limitations of previous-generation cloud remote sensing. I was especially surprised – going from the introduction to the methods – the lack of connection (and motivation) to previous spaceborne remote sensing studies using (for example) CloudSat measurements and why CloudCube will be an improvement over a long period of time. I have listed several reading recommendations in my specific comments below – and I encourage the authors to review embedded references within these references to strengthen the introduction further. In addition, I've recommended at several points to either comment on or carry out some additional analysis to better connect your work with the existing literature (the paper itself already has enough publishable results, but I will leave it to

the authors to decide the best course of action as I do believe some of these small-but-important details are necessary to strengthen the manuscript).

There are also numerous places in the text (see specific comments below, especially after about ~L120 or so) where the authors need to be more quantitative. In most places, saying "small differences" or "small percentage" (among other similar phrases) is technically correct – but values need to be provided so the reader is not misled in any way. This is especially true for relative humidity or any variable involving moisture.

We have addressed this issue in the new version of the manuscript.

**Specific/Technical Comments**

L37-39: Sub-cloud evaporation should probably be mentioned here too.

Kalmus, P., & Lebsock, M. (2017). Correcting biased evaporation in CloudSat warm rain. IEEE Transactions on Geoscience and Remote Sensing, 55(11), 6207-6217.

Evaporation of drizzle beneath cloud base is indeed an important thermodynamic influence on the boundary layer structure and therefore cloud amount and extent. We chose not to bring up this detail in the introduction however since there are many processes that similarly influence the clouds including at least cloud top entrainment, radiative cooling, precipitation initiation, aerosol scavenging, etc.

L39: Given this manuscript's connection to CloudCube, it is very important (in my view) to mention limitations of previous satellite-based remote sensing efforts – especially with measuring thin clouds in the lowest 1-km above ground.

Stephens, G. L., et al. (2008), CloudSat mission: Performance and early science after the first year of operation, J. Geophys. Res., 113, D00A18, doi:10.1029/2008JD009982.

This is a great point. In fact, in a follow-up paper, we plan to explore the possibility of using similar spaceborne measurements given realistic spaceborne radar performance characteristics. The focus of this paper is on ground-based methods to retrieve the LWC profile. For the foreseeable future, we will not have dual-frequency satellite observations with the sensitivity to detect cloud water backscattering signals, so we want to focus on ground-based in this paper where we feel there is a good opportunity to advance observational capability over the coming years.

L45-47: Can you provide some sort of numeric example explaining what value(s) of differential attenuation correspond to value(s) of LWC? This won't be obvious to your casual reader.

The differential attenuation that corresponds to a value of LWC depends on the frequency of operation of the radar, the frequency separation between the two radar frequencies, the cloud properties, etc. We believe that it is complicated to provide such example without misleading the reader.

L51: You explain this more in the next sentence, but this statement should be concluded with relevant citations (i.e., what studies "suggest the use of high frequencies and large frequency pair separations?).

We have added the following citations in line 53:

Hogan, R. J., Gaussiat, N. and Illingworth, A. J.: Stratocumulus liquid water content from dual-wavelength radar, J. Atmos. O. Tech., 22(8), 1207-1218. https://doi.org/10.1175/JTECH1768.1, 2005.

Williams, J. K., and Vivekanandan, J.: Sources of error in dual-wavelength radar remote sensing of cloud liquid water content. J. Atmos. O. Tech., 24(8), 1317-1336. https://doi.org/10.1175/jtech2042.1, 2007.

L54-61: This is an important statement, but to this point of the introduction, there is limited context for other multi-platform techniques that attempt to quantify cloud liquid (or rain) water. The introduction is especially limited in context from other spaceborne cloud remote sensing studies given this manuscript's connection to CloudCube.

CloudSat + MODIS warm rain retrieval algorithm:

Lebsock, M. D., & L'Ecuyer, T. S. (2011). The retrieval of warm rain from CloudSat. Journal of Geophysical Research: Atmospheres, 116(D20).

L'Ecuyer, T. S., and G. L. Stephens (2002), An estimation-based precipitation retrieval algorithm for attenuating radars, J. Appl. Meteorol., 41, 272–285, doi:10.1175/1520-0450(2002)041<0272:AEBPRA>2.0.CO;2.

Airborne radar & radiometric retrievals of cloud and rain water designed for thin stratocumulus in the SE Atlantic Ocean:

Dzambo, A. M., L'Ecuyer, T., Sinclair, K., van Diedenhoven, B., Gupta, S., McFarquhar, G., O'Brien, J. R., Cairns, B., Wasilewski, A. P., and Alexandrov, M.: Joint cloud water path and rainwater path retrievals from airborne ORACLES observations, Atmos. Chem. Phys., 21, 5513–5532, https://doi.org/10.5194/acp-21-5513-2021

Once again, we chose to just focus on the ground-based retrievals in this manuscript.

L57: Given the scope of this manuscript, I highly recommend expanding your introduction here to discuss triple-frequency radar observations, and discuss how/why a G-band/Ka-band retrieval is optimal for thin cloud (this will be obvious at this point to most readers, but not everyone will be aware of triple-frequency radar algorithm technology).

Battaglia, A., Tanelli, S., Tridon, F., Kneifel, S., Leinonen, J., & Kollias, P. (2020). Triple-frequency radar retrievals. Satellite Precipitation Measurement: Volume 1, 211-229.

Triple-frequency techniques have primarily been used to probe the microphysics of frozen hydrometeors. We feel that adding these references would prove distracting to uninformed readers as we are working here with liquid-phase hydrometeors where the density is constrained.

L71-72: To be clear, your study is *not* a statistical analysis of the whole EPCAPE dataset, but rather a single case study based on 100 continuous minutes of data? Please clarify. This information would be more appropriate at the beginning of Section 2.

We have clarified this in line 95-99:

"We have selected two case studies in two different days for the analysis developed in this work. The first case, recorded on 17 April 2023, at 21:19:00 UTC starting time, lasts 100 minutes while the second data set, taken on 25 April 2023, at 13:12:00 UTC starting time, continues for 20 minutes. The radar data that have been used for the LWC retrieval consist of close to 18000 and 1800 profiles of observed echo power at G-band and Ka-band, respectively, which have been averaged over 60 s to retrieve 120 LWC profiles."

L90: Consider one of two things in your next version of this paper, either (1) comparing your results (at an appropriate point later in the paper) against those produced by the classic Hitschfeld and Borden (1954) attenuation correction scheme, which is widely used and known, or (2) commenting on how your approach might compare against the classic Hitschfeld and Borden attenuation correction. Most studies that I'm aware of on the topic of radar attenuation use Hitschfeld and Borden – it would be prudent (for visibility sake) that you not neglect this detail.

Hitschfeld, W., & Bordan, J. (1954). Errors inherent in the radar measurement of rainfall at attenuating wavelengths. Journal of the Atmospheric Sciences, 11(1), 58-67.

We have added the following paragraph in lines 302-307 and added the Hitschfeld and Borden reference:

"An alternative method to account for the radar attenuation is based on the Hitschfeld and Borden (1954) scheme, where the reflectivity would be corrected at each range gate prior to computing the LWC from a reflectivity-based relationship. For a range resolution of 30 m and LWC smaller than 1 $gm^{-3}$, the attenuation by cloud liquid water at Ka-band is expected to be on the order of 0.01 dB per range gate as computed using ITU (2013). Even over the depth of a 300 m cloud, the cumulative attenuation at Ka-band would be on the order of 0.1 dB. Therefore, the reflectivity-based retrieval is not significantly affected by attenuation in this particular scene."

L118-121: This section can probably be merged with the previous paragraph. Also, "... deduced from the ceilometer cloud base measurement" is somewhat confusing – do you mean to say if the ceilometer detects a cloud base near the surface? I would explain this further.

We have merge both paragraphs and modified the sentence (line 122):

"… we have excluded regions below the cloud base measured by the ceilometer."

L126: I presume these "small" differences also accounted for diurnal variability? I would quantify what a "small" difference is in temperature, humidity, etc.

We have added the following correction in line 128:

"…less than 0.5 K, 7 hPa, and 3%, respectively"

L127: Again, quantify what this "small percentage" is.

We have added the following correction in line 130-131:

"Even if there had been a variation of 1 K, … would be smaller than 0.03 $gm^{-3}$…"

L130-134: This information needs to be at the beginning of Section 2, so the reader is clear about what data from what times are used in this analysis. This will also improve the flow of your Section 2 here, as this is awkwardly placed here.

We have moved and provided more clarity in this paragraph (line 96-102).

L146: Specify in the caption what positive/negative Doppler velocity corresponds to (i.e., toward or away from the radar).

We have added the following sentence in line 147-148:

"Negative Doppler velocities correspond to particles moving towards the radar."

L159: What is the value of the "uncertainty in the radar reflectivity measurement"? Quantify please.

As shown in Eq. (2), the value of the uncertainty in the radar reflectivity measurement depends on multiple radar parameters. We believe that providing values for particular radar configurations can be confusing for the reader.

L164: "huge leap" is a bit informal, and could mean different things to different readers. It may be worth mentioning here (please check in the literature to be sure) that a Ka-band/G-band technique *may* be the widest frequency differential for a technique of this kind (and this will make the novelty of your work more obvious to your reader!).

The recommendation has been accepted (line 159-160).

L221: what exactly is a "short profile"?

We have changed the sentence in line 218:

"We fit a portion of the DFR(r) profile…"

L300: You can abbreviate liquid water content to LWC.

The recommendation has been accepted (line 301).

L416-423: These are great results – some of these number should be weaved into your abstract to make it more quantitative.

The recommendation has been accepted.

Summary and Conclusions: I have no issues with your algorithm and results and maintain that this work is novel and should be published as soon as possible... however, readers who are familiar with radiometric + radar-based retrievals of cloud LWC may not totally be convinced your work/algorithm is a significant advancement over existing radiometric + radar-based algorithm technology. It would be overkill to add this to your paper, but you should definitely mention in this section as a "future work" that you intend to compare your results against an optimized radiometric + radar joint algorithm to see which one performs better. In particular, a radiometric observation of total optical depth was valuable in constraining the *total* liquid water path of the cloud, and in

combination with radar data, can also be used to constrain rainwater contents (see the aforementioned Lebsock and L'Ecuyer, 2011 and Dzambo et al. 2021 references). Again, this would be overkill for this study, but your summary paragraph here should include at least a couple sentences (perhaps a full paragraph) discussing these caveats and avenue of future work.

This section should also include a statement (1-2 sentences) that this work was based on a single set of continuous (100-minute) measurements, and that the robustness of your work will be tested when more data (or similar datasets from other campaigns) are made available.

As an aside – I would really like to see you and your group carry out this suggested follow-on study using radiometric + radar-based retrievals of LWC from EPCAPE!

We have added the following paragraph (line 492-498):

"Future work will be devoted to testing, improving, and validating the results retrieved in this work. KAZR and CloudCubeG will be participating in the Cloud and Precipitation Experiment at Kennaook (CAPE-k), where close to one-year worth of data will be available by the end of 2025. At CAPE-k, KAZR and CloudCubeG will be accompanied by the W-band ARM Cloud Radar (WACR), and we will exploit the availability of a triple-frequency radar system to benefit the LWC retrieval as described in Sect. 7. Finally, we intend to compare the results against a radiometer and single-frequency radar joint retrieval, as combining the radiometric observation of total optical depth with the radar capability of profiling can also be used to constrain the cloud LWC."

L449: "indicative of their radiative properties" – LWC is not the *only* property of clouds that matter! Please expand/rephrase to make this more accurate.

We have modified the sentence in line 458:

"…is key to better understand"

L481: I'd say "atmosphere" rather than "medium".

The recommendation has been accepted (line 491).